

# Flower structure and development in Vietnamese *Sciaphila* (Triuridaceae: Pandanales): refined knowledge of the morphological misfit family and implications for taxonomy

Maxim S. Nuraliev[1,2], Margarita V. Remizowa[1] and Dmitry D. Sokoloff[1]

[1] Department of Higher Plants, Biological Faculty, M.V. Lomonosov Moscow State University, Moscow, Russia
[2] Joint Russian-Vietnamese Tropical Scientific and Technological Center, Hanoi, Vietnam

Corresponding author
Maxim S. Nuraliev,
max.nuraliev@gmail.com

## ABSTRACT

The monocot family Triuridaceae is a morphological misfit with respect to several traits of floral morphology, including the uniformly apocarpous polymerous gynoecium and the famous inside-out flowers of *Lacandonia*. Although Triuridaceae are crucially important for understanding the floral evolution of Pandanales and angiosperms in general, significant knowledge gaps exist which hamper adequate morphological analysis of flowers in this family. The scarcity of morphological data is also reflected in numerous taxonomic inconsistencies. Here we provide a comprehensive study of reproductive organs of four species of *Sciaphila* occurring in Vietnam (*S. arfakiana*, *S. densiflora*, *S. nana* and *S. stellata*) including the first investigation of early floral development and gynoecium phyllotaxis. Our observations are mainly based on SEM images. We confirm the perianth (studied in male flowers) to be two-whorled and report a rare sequence of initiation of perianth parts: the outer tepals show a late congenital fusion, as their free lobes appear before the common perianth tube, whereas the inner tepals show an early congenital fusion, with their free lobes initiating on the tube rim. We interpret the stamen appendages as basal adaxial outgrowths of the stamen filaments. We discuss the number of thecae and locules in anthers of *Sciaphila*, and conclude that 3- and 4-, but not 2-locular anthers are characteristic of this genus. We describe the carpels as consisting of both ascidiate and plicate zones, the former being extremely short and completely obscured by anthesis. The single ovule is attached in the cross-zone. The style is non-plicate. We analyze gynoecium phyllotaxis by estimating its contact parastichies, and by investigating the number and arrangement of the outermost carpels. The carpel arrangement in a given gynoecium is a result of the balance between whorled and irregular (but not spiral) phyllotaxis. We recognize the following figures of gynoecium merism: 6, 9, 10, 10.5, 11 and 12, with the prevalence of those divisible by three. We discuss our results in the light of general diversity of floral structure of monocots. Our data allow to clarify several issues in taxonomy of Asian *Sciaphila* and indicate directions of further studies. We report a significant range extension for *S. densiflora*, which is newly recorded for the flora of Vietnam. We describe for the first time staminodes in female flowers of this species. We reveal

two distinct morphs of *S. nana* in Vietnam. We highlight the need of a thorough revision of *S. secundiflora* species complex in order to verify the species boundaries and, in particular, to test the identity of the Vietnamese *S. stellata*.

# INTRODUCTION

The mycoheterotrophic family Triuridaceae (Pandanales) is remarkable for the outstanding diversity of reproductive (stamen- and carpel-bearing) units. Since the discovery of this family, reproductive units of Triuridaceae have attracted close attention of evolutionary morphologists, which resulted in emergence of various and conflicting hypotheses on their morphological interpretation. In particular, it has been argued that the reproductive units are perianth-bearing flowers (*Vergara-Silva et al., 2003*; *Ambrose et al., 2006*; *Álvarez-Buylla et al., 2010*; *Espinosa-Matías et al., 2012*) or intermediate structures between flowers and pseudanthia (*Rudall, 2003*, *2008*; *Rudall & Bateman, 2006*). The floral hypothesis is currently widely accepted (see e.g. *Rudall, Alves & Sajo, 2016*) and followed in this paper. The other extensively discussed issue is the homology of filamentous structures that are found in various floral organs across Triuridaceae. These structures include a column-like organ in the floral center, interstaminal organs and the appendages of tepals and stamens. *Rudall (2008)* largely interpreted them as morphological novelties (and organs *sui generis*) based on their similarity to each other in shape and late appearance in flower development. However, they are more commonly assumed to represent modified elements (or their parts) of perianth and androecium (*Maas-van de Kamer & Weustenfeld, 1998*; *Ambrose et al., 2006*; *Merckx et al., 2013*).

The flowers of Triuridaceae are unusual for monocots in a number of traits. They are more commonly unisexual (but sometimes bisexual or only functionally unisexual), with a perianth of usually three, four or six (but up to 10) tepals basally united in a common tube, androecium of typically two, three or six stamens that are united in an androphore or free from each other, and a gynoecium of numerous free one- or two-ovuled carpels (*Maas & Rübsamen, 1986*; *Rübsamen-Weustenfeld, 1991*; *Maas-van de Kamer & Weustenfeld, 1998*; *Rudall, 2008*; *Merckx et al., 2013*).

Despite considerable progress in understanding of the reproductive structures of Triuridaceae achieved in recent decades, information on floral development and sometimes even floral groundplan is still lacking or incomplete for many taxa of the family. These gaps in knowledge cause numerous persisting taxonomic problems, and also hamper investigations of floral evolution in this family. The main uncertainties are the following. Whereas the perianth of Triuridaceae with three tepals is apparently single-whorled, in four-tepaled and six-tepaled representatives of the family it is described as either single-whorled or two-whorled (*Rübsamen-Weustenfeld, 1991*; *Maas-van de Kamer, 1995*; *Gandolfo, Nixon & Crepet, 2002*). The structure of the androecium remains unclear in some species because of the questionable morphological nature of the staminode-like

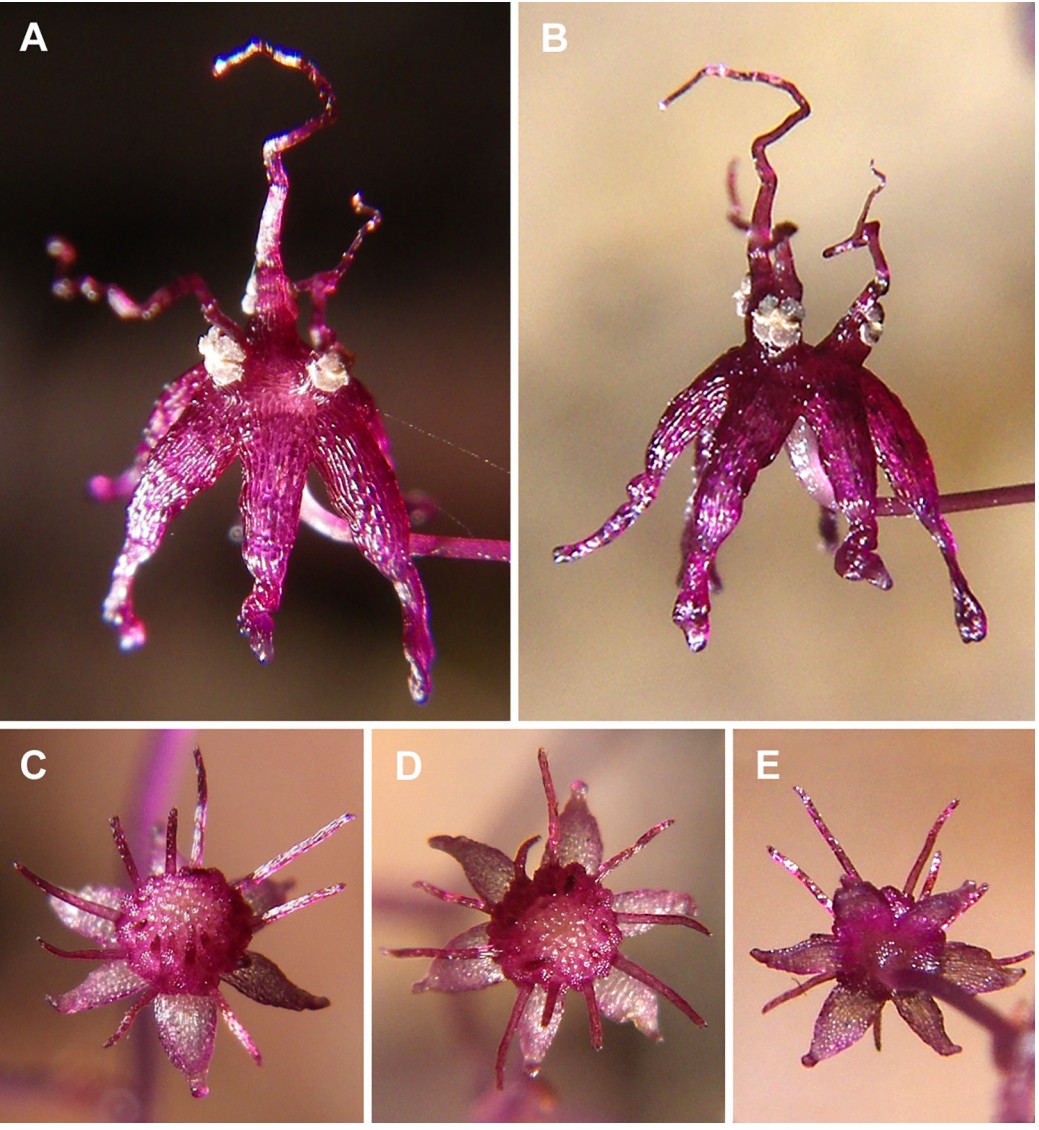

**Figure 1** *Sciaphila arfakiana* (*Nuraliev et al. 2499*). (A and B) Male flower, side view. (C and D) Female flower at early anthesis, oblique and top view. (E) Female flower, view from below. Photos: Maxim Nuraliev.

filamentous organs (*Rudall, 2008*), uncertain nature of the stamen appendages and apparently variable number of anther locules (= microsporangia) (*Van de Meerendonk, 1984*). The carpels of Triuridaceae have never been consistently described in terms of their zonation (i.e., the occurrence of a plicate and/or ascidiate zone) and type of placentation. Finally, gynoecium phyllotaxis, which is extremely diverse and complicated in this family, has been so far evaluated only for a few representatives (*Rudall, 2008*).

In this article, we investigate floral structure and development in the genus *Sciaphila* Blume, exemplified by four Asian species: *Sciaphila arfakiana* Becc. (Fig. 1), *S. densiflora* Schltr. (Fig. 2), *S. nana* Blume (Figs. 3 and 4) and *S. stellata* Aver. (Fig. 5). The androecium

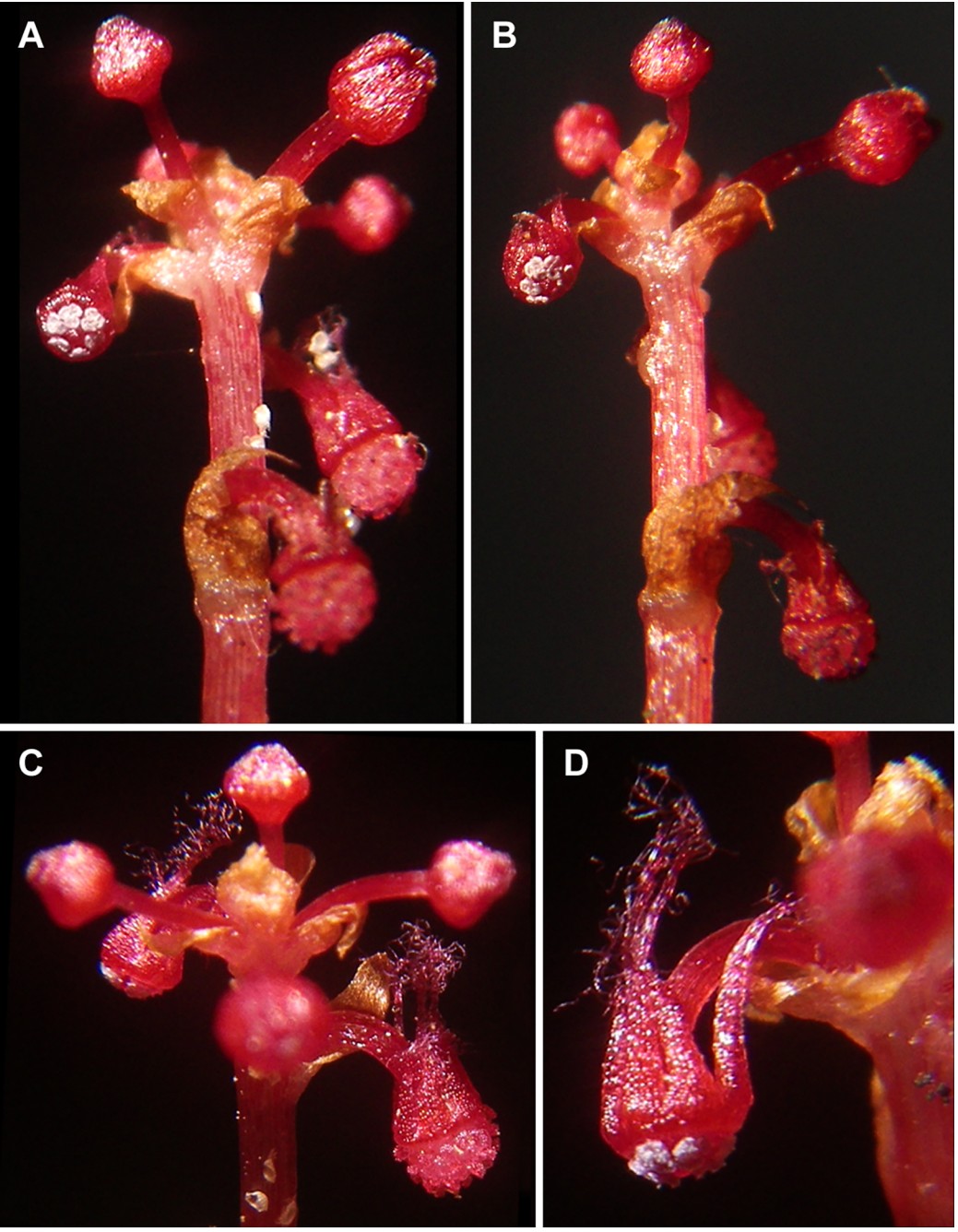

**Figure 2 *Sciaphila densiflora* (*Nuraliev 1670*).** (A and B) Inflorescence, showing male flowers in the distal part and female flowers in the proximal part. (C) Part of inflorescence, showing details of female flower. (D) Male flower, side view. Photos: Maxim Nuraliev.

is the most variable part of the floral groundplan of these species: there are usually six stamens in male flowers of *S. densiflora* and three stamens in those of the other species (*Van de Meerendonk, 1984*; *Averyanov, 2007*). We discuss our results in the context of the floral diversity of the entire family and particularly compare them with available data

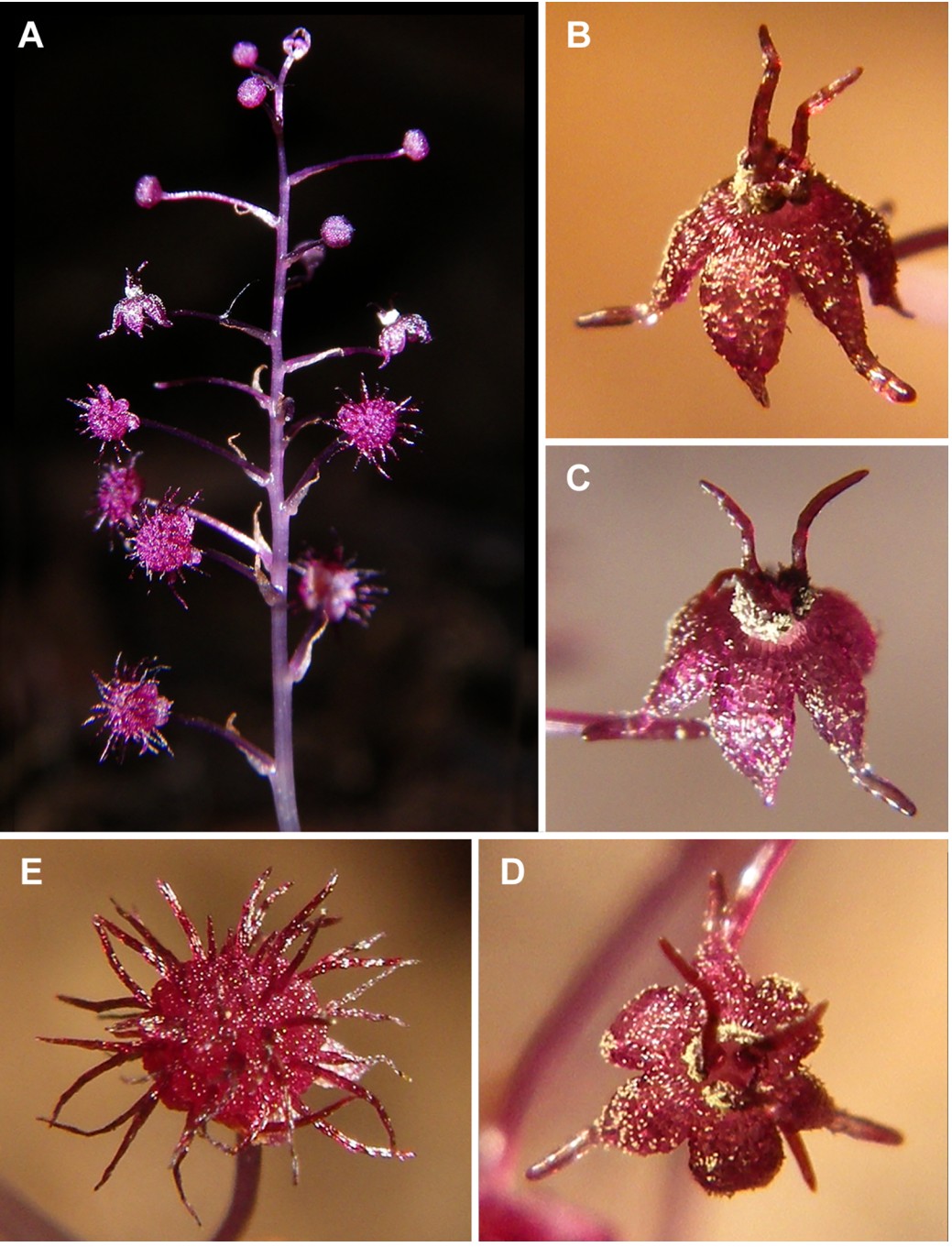

**Figure 3** *Sciaphila nana* (*Nuraliev 2445*). (A) Inflorescence, showing male flowers in the distal part and female flowers in the proximal part. (B and C) Male flower, oblique view. (D) Male flower, top view. (E) Female flower at late anthesis, top view. Photos: Maxim Nuraliev.

on other species of *Sciaphila* and also *Seychellaria* Hemsl., according to their high morphological similarity and a nested position of *Seychellaria* within *Sciaphila* in the molecular phylogenetic reconstructions (*Mennes et al., 2013*). Keeping in mind that

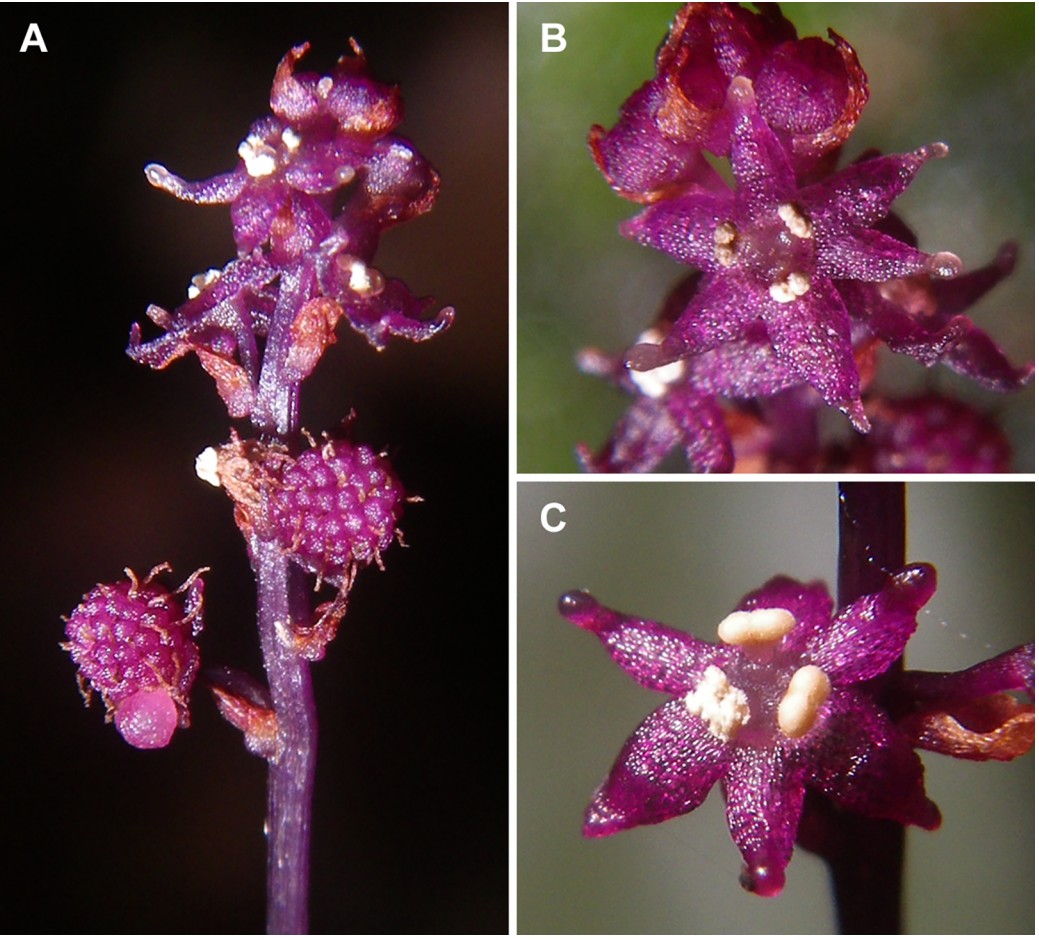

**Figure 4 *Sciaphila nana* (*Nuraliev et al. 972*).** (A) Inflorescence, showing male flowers in the distal part and female flowers (or young fruits) in the proximal part. (B and C) Male flower, top and oblique view. Photos: Maxim Nuraliev.

merging of these two genera under the name *Sciaphila* (as proposed by *Christenhusz, Fay & Byng (2018)*) is likely to be reasonable, we prefer to treat them separately until more evidence is available, including a phylogenetic investigation based on a broader sampling. *Sciaphila* and *Seychellaria* belong to Sciaphileae, the only Asian tribe of Triuridaceae; two other tribes, Kupeaeae and Triurideae, are endemic to tropical Africa and the Neotropics correspondingly (*Cheek, 2003*; *Mennes et al., 2013*; *Merckx et al., 2013*). Our specific goals were: (1) to uncover patterns of early development of the perianth and to establish the sequence of initiation of perianth parts; (2) to document stages of development of individual carpels, to define the structural zones of carpel of *Sciaphila* and to homologize it with those of other angiosperms; (3) to investigate the phyllotaxis patterns in female flowers and to reveal modes of carpel arrangement and gynoecium merism for the cases of whorled arrangement; (4) to amend morphological characteristics of some of the species with respect to such features as the number of anther locules and presence of staminodes in female flowers; (5) to reassess taxonomy and geography of the studied species in the light of the new morphological data.

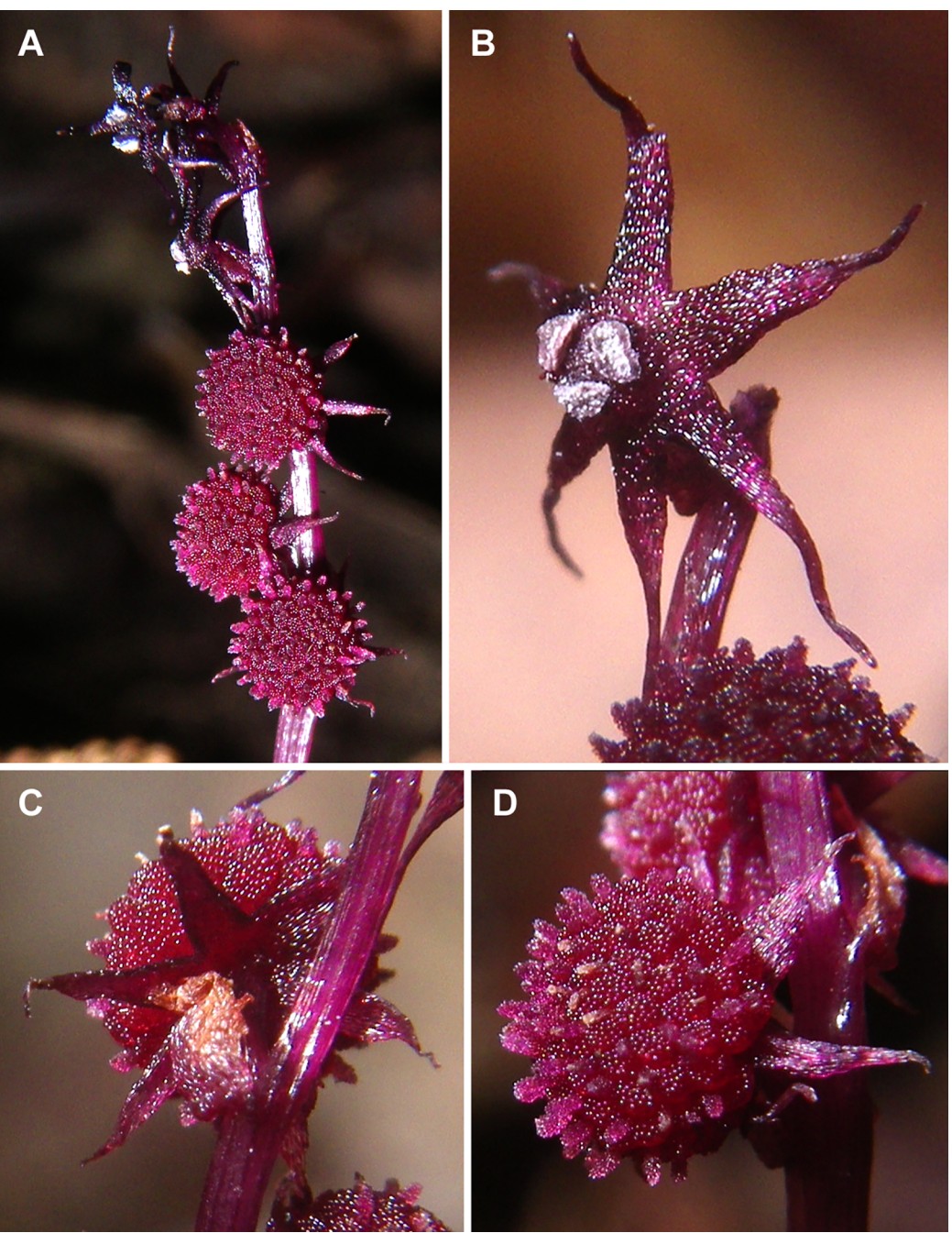

**Figure 5** *Sciaphila stellata* (*Nuraliev & Kuznetsova 1380b*). (A) Inflorescence, showing male flowers in the distal part and female flowers in the proximal part. (B) Male flower, oblique view. (C) Female flower, view from below. (D) Female flower, side view. Photos: Maxim Nuraliev.

## MATERIALS AND METHODS

The information on studied specimens is given in Table 1. The material was collected during expeditions of the Russian-Vietnamese Tropical Centre under permits from the Vietnamese Government with participation of the Russian-Vietnamese Tropical Center

**Table 1 Voucher specimens for the examined material.**

| Species | Voucher | Location and date of collection |
|---|---|---|
| *Sciaphila arfakiana* Becc. | *M.S. Nuraliev 1616* (MW) | Vietnam, Gia Lai Province, K'Bang District, Son Lang Municipality, Kon Chu Rang Nature Reserve, 29 Km ESE of Mang Den town, forest, near river, N 14°31′19″, E 108°32′55″, 930 m a.s.l., 01.06.2016 |
| | *M.S. Nuraliev, A.N. Kuznetsov, S.P. Kuznetsova 2499* (MW) | Vietnam, Quang Nam Province, Nam Giang District, Song Thanh Nature Reserve, forest on slope, N 15°33′58″, E 107°23′16″, 1,030 m a.s.l., 05.05.2019 |
| *Sciaphila densiflora* Schltr. | *M.S. Nuraliev 1670* (MW) | Vietnam, Kon Tum Province, Kon Plong District, Thach Nham protected forest, 17 Km N of Mang Den town, forest on slope, N 14°45′00″, E 108°18′15″, 1,200 m a.s.l., 08.06.2016 |
| *Sciaphila nana* Blume | *M.S. Nuraliev 498* (MW) | Vietnam, Dak Lak Province, Lak District, Bong Krang Municipality, Chu Yang Sin National Park, 14 Km S of Krong Kmar village, forest, in ravine, not far from stream, N 12°22′41″, E 108°21′11″, 1,640 m a.s.l., 03.04.2012 |
| | *M.S. Nuraliev, A.N. Kuznetsov, S.P. Kuznetsova 972* (MW) | Vietnam, Dak Lak Province, Lak District, Bong Krang Municipality, Chu Yang Sin National Park, 12 Km SSE of Krong Kmar village, mixed forest, N 12°24′29″, E 108°23′15″, 1,530 m a.s.l., 25.05.2014 |
| | *M.S. Nuraliev, S.P. Kuznetsova 1380a* (MW) | Vietnam, Kon Tum Province, Kon Plong District, Thach Nham protected forest, 17 Km N of Mang Den town, forest on slope, N 14°45′00″, E 108°18′15″, 1,200 m a.s.l., 20.04.2015 |
| | *M.S. Nuraliev 1561* (MW) | Vietnam, Gia Lai Province, K'Bang District, Son Lang Municipality, Kon Chu Rang Nature Reserve, 29 Km ESE of Mang Den town, forest, near river bank, N 14°30′50″, E 108°32′45″, 1,000 m a.s.l., 26.05.2016 |
| | *M.S. Nuraliev 1669* (MW) | Vietnam, Kon Tum Province, Kon Plong District, Thach Nham protected forest, 17 Km N of Mang Den town, forest on slope, N 14°45′00″, E 108°18′15″, 1,200 m a.s.l., 06-08.06.2016 |
| | *M.S. Nuraliev 2445* (MW) | Vietnam, Quang Nam Province, Nam Giang District, Song Thanh Nature Reserve, forest, river bank, N 15°34′12″, E 107°22′39″, 1,050 m a.s.l., 30.04.2019 |
| *Sciaphila stellata* Aver. | *M.S. Nuraliev, S.P. Kuznetsova 1380b* (MW) | Vietnam, Kon Tum Province, Kon Plong District, Thach Nham protected forest, 17 Km N of Mang Den town, forest on slope, N 14°45′00″, E 108°18′15″, 1,200 m a.s.l., 20.04.2015 |
| | *M.S. Nuraliev, A.N. Kuznetsov, S.P. Kuznetsova 2499bis* (MW) | Vietnam, Quang Nam Province, Nam Giang District, Song Thanh Nature Reserve, forest on slope, N 15°33′58″, E 107°23′16″, 1,030 m a.s.l., 05.05.2019 |

(permit numbers: 308, 547, 1951). The precise locations of the specimens can be seen on a map provided by *Nuraliev et al. (2019)*. The whole plants and inflorescences were fixed and stored in 70% ethanol. The photographs of living plants were taken with a Pentax Optio W80 digital camera (Pentax Corporation, Tokyo, Japan).

For scanning electron microscopy (SEM), at Moscow State University, the inflorescences and flowers were dehydrated in 96% ethanol followed by 100% acetone. Dehydrated material was critical-point dried using a HCP-2 critical point dryer (Hitachi, Tokyo, Japan), coated with gold and palladium using an Eiko IB-3 ion-coater (Eiko Engineering Co. Ltd., Tokyo, Japan), and observed using a CamScan 4 DV (CamScan, Cambridge, UK) SEM at Moscow State University.

All of the images were treated using PHOTOSHOP ELEMENTS (Adobe Systems, San Jose, CA, USA). Some SEM images were coloured using CORELDRAW X5 (Corel Corporation, Ottawa, ON, Canada).

# RESULTS: FLORAL DEVELOPMENT AND MORPHOLOGY

The basic inflorescence unit of all examined species is a raceme with female flowers in the proximal part and male flowers in the distal part. Each flower has a subtending bract and

lacks floral prophylls (bracteoles). The floral development of the four species studied here is essentially uniform. Below we provide a generalized description which is based on all the obtained data, and highlight the differences between studied species and specimens. For *Sciaphila stellata*, only late developmental stages were studied.

**Male flowers (Figs. 6–18)**

*Flower initiation*
The floral primordium arises in the axil of its subtending bract, soon after the appearance of the flower-subtending bract on the inflorescence apex. The floral primordium is narrowly elliptic (much wider in the transversal plane than in the median plane), ca. 100 μm × 30 μm in *S. arfakiana* (Figs. 6 and 7A) and *S. nana* (Figs. 13A and 13B) and nearly twice smaller in *S. densiflora* (Fig. 10A). The flower-subtending bract is only slightly wider and higher than the floral primordium and thus does not cover the young flower in early development. Apparently soon after this stage (as it is usually seen in the previous flower of a raceme), the young flower obtains a nearly globose shape, and the flower-subtending bract elongates significantly and completely encloses the developing flower.

Prior to the initiation of the first floral elements, the developing flower (as observed in *S. arfakiana*, Fig. 6A, and *S. nana*, Figs. 13A and 13B) has an isosceles triangular shape (in top view) with a slightly convex upper surface. Flowers of this structure are usually the third or fourth flowers visible in the raceme (counting from the apex); their width is about 120 μm in *S. arfakiana* and *S. nana*.

*Initiation and early development of perianth*
The floral organs initiate in an acropetal sequence. Three tepals, one occupying the median abaxial position and two in transversal-adaxial positions, are the first organs to become discernible (Figs. 6 and 13). They initially can be recognized as short triangular outgrowths around a large and convex floral apex. Although the three tepals arise more or less simultaneously, the median abaxial tepal (the one closest to the flower-subtending bract) is considerably smaller than the two others since the earliest stages and during some further development. The appearance of the entire flower is thus prominently monosymmetric at the corresponding stages. Soon after the initiation of the three first-formed tepals, a low but distinct rim differentiates around the flower periphery (Figs. 13D and 14A). This rim connects the bases of the three tepals, and therefore we interpret it as a perianth tube. Then, three other tepals initiate alternating with the first three tepals (i.e., one in a median adaxial position and two in transversal-abaxial positions). Their primordia arise as outgrowths of the perianth tube (Figs. 6, 7, 10B, 10C and 14A–14D). The median adaxial tepal initiates simultaneously with or slightly earlier (Fig. 7B) than the two transversal-abaxial tepals. The establishment of the tepals and the perianth tube apparently takes place during quite a short period of time; when the flower acquires all these structures, its diameter does not exceed 200(–250) μm in *S. arfakiana* and *S. nana* and ca. 200 μm in *S. densiflora*.

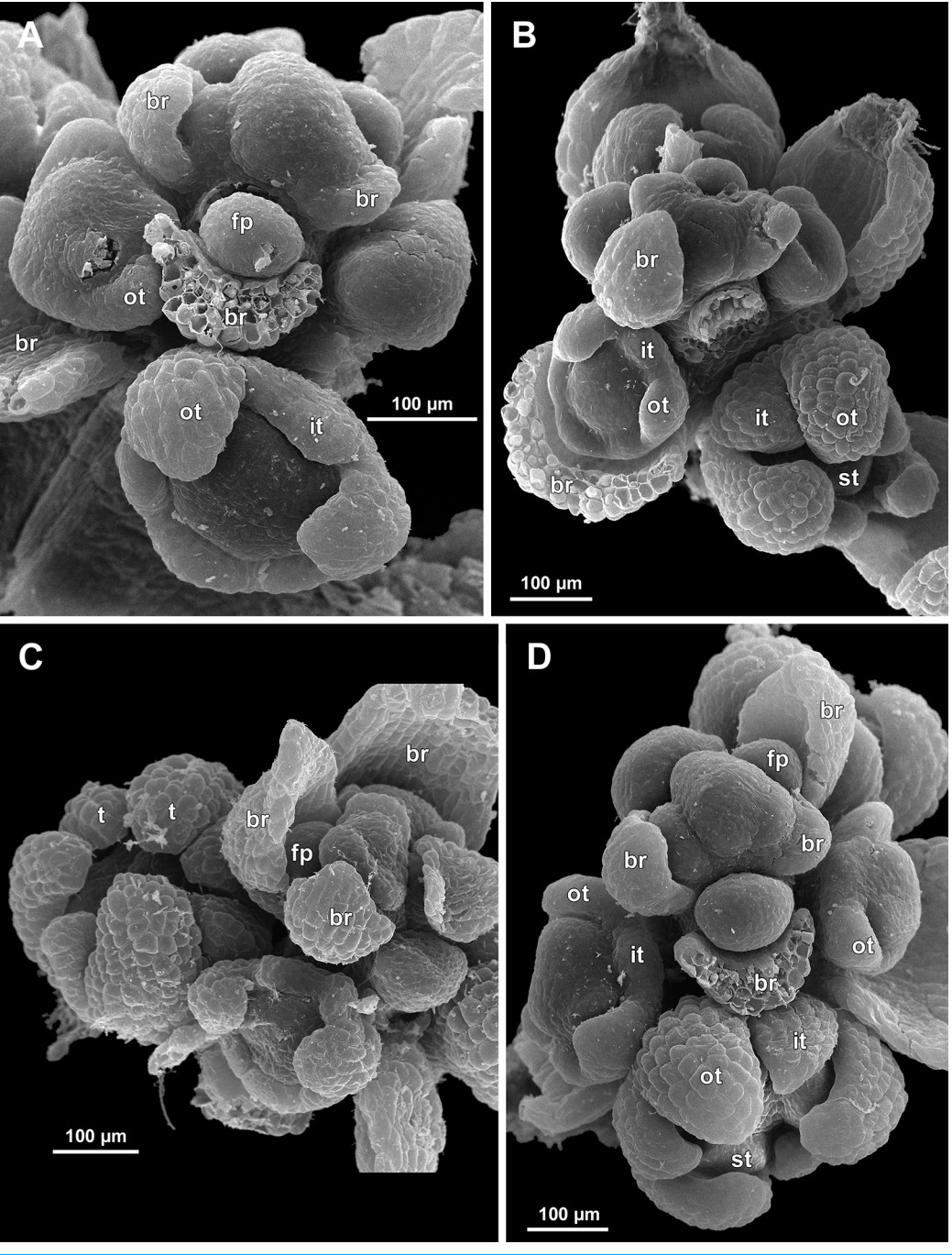

**Figure 6** *Sciaphila arfakiana*, **distal parts of young inflorescences, showing early development of male flowers (SEM) (A, C and D: *Nuraliev 1616*, B: *Nuraliev et al. 2499*).** Note a flower with seven tepals in C. Labels: br, flower-subtending bract; fp, flower primordium; it, inner tepal; ot, outer tepal; st, stamen; t, tepal.           

  The perianth groundplan was occasionally different from the typical condition with six tepals. An inflorescence of *S. arfakiana* was found bearing two flowers with seven tepals each at the stage of perianth development (along with several other flowers) (Fig. 6C). In one of these flowers, two tepal lobes were larger (probably early-arising) and five lobes

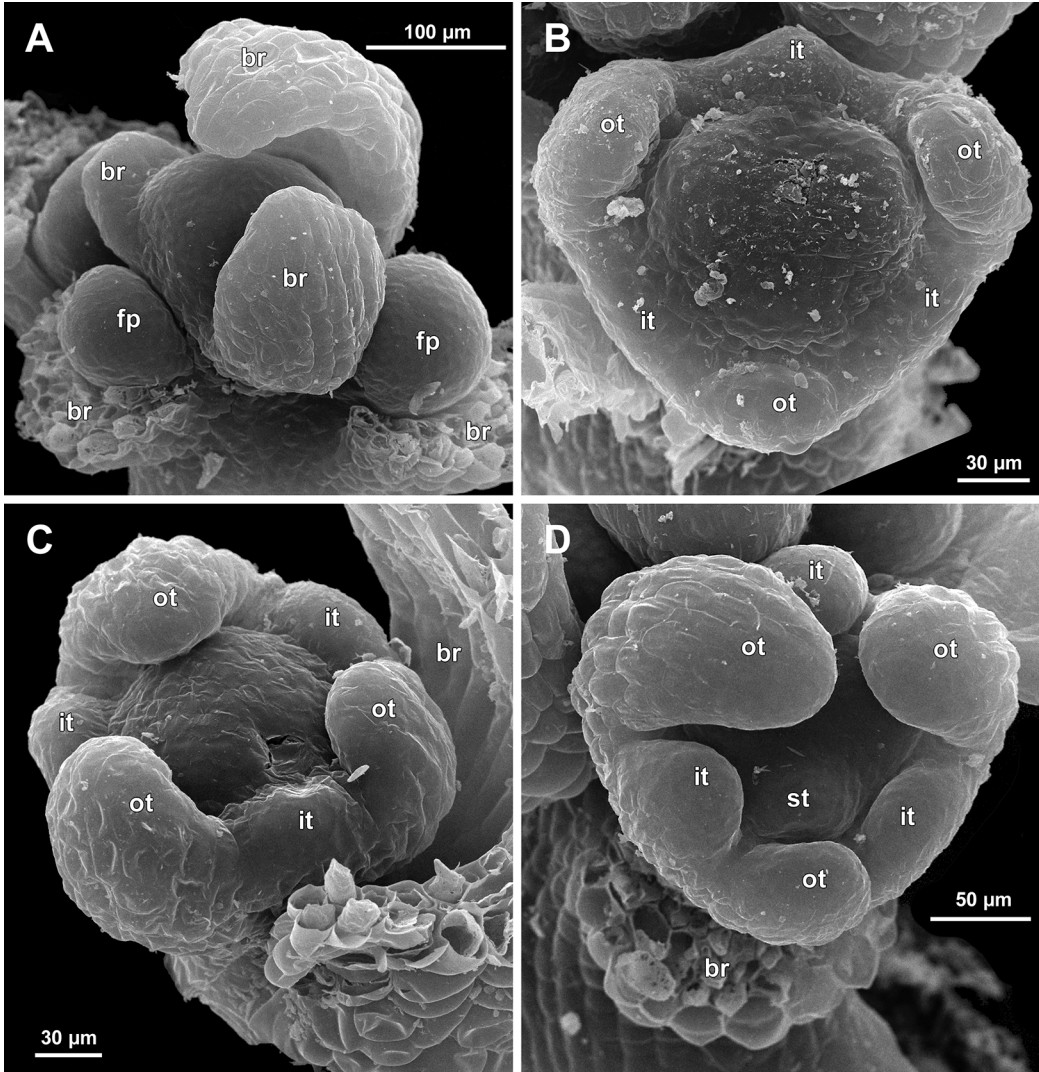

**Figure 7** *Sciaphila arfakiana*, **early development of male flowers (SEM) (A, C, D: *Nuraliev et al. 2499*, B: *Nuraliev 1616*).** (A) Distal part of inflorescence, showing flower primordia. (B) Flower at initiation of free lobes of inner tepals, top view with abaxial side facing down. (C) flower at perianth development, oblique view. (D) Flower at stamen initiation, top view. Labels: br, flower-subtending bract; fp, flower primordium; it, inner tepal; ot, outer tepal; st, stamen.

were much smaller (probably late-arising); the other flower possessed three large and four small tepals. Out of many examined flowers of *S. nana*, three flowers possessed four tepals. One of them had four stamens (instead of the typical condition of three stamens, see below) (Fig. 15B), and for the others the stamen number is unknown (e.g., Fig. 13B).

*Flower at initiation and early development of androecium*
There is a very short plastochron between the initiation of the second series of tepals and the stamens (Figs. 6, 7, 10B, 13A and 14B–14D). In some flowers, the stamen primordia are clearly visible even when the three late-arising free lobes of tepals are still weakly pronounced (but the perianth tube is already formed). This is especially evident in
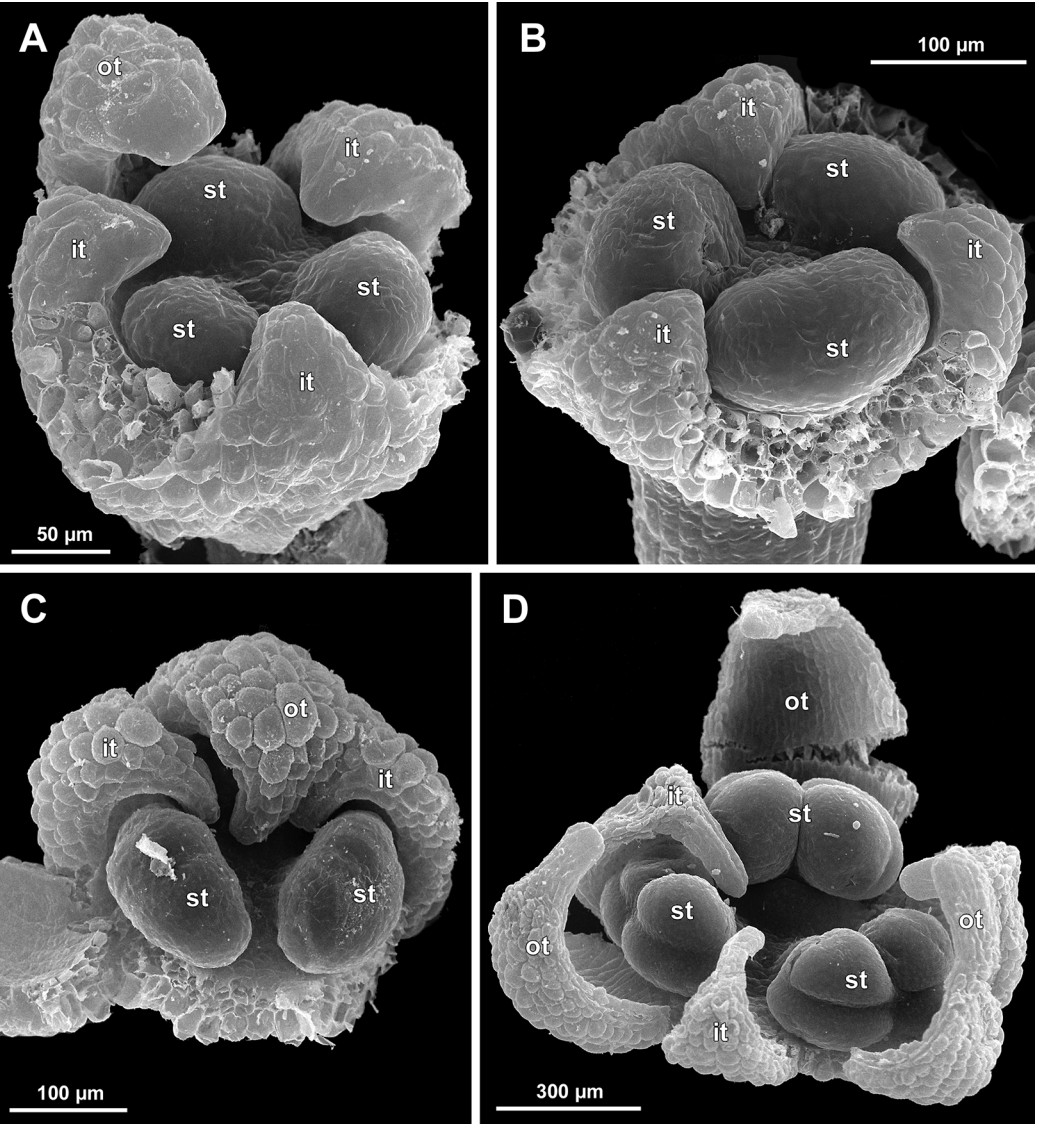

**Figure 8** *Sciaphila arfakiana*, **development of male flowers (SEM) (A–C: *Nuraliev 1616*, D: *Nuraliev et al. 2499*).** (A) Flower with young stamens, oblique view (two outer tepals removed). (B) Flower at anther differentiation, oblique view (outer tepals removed). (C) Flower with tepals covering stamens, oblique view (two outer and one inner tepals removed). (D) Flower with differentiated anther locules, oblique view (one inner tepal removed). Labels: it, inner tepal; ot, outer tepal; st, stamen.

*S. densiflora*, where three of the six stamens are strongly prominent and the late-arising free lobes of tepals are hardly discernible at a certain stage of flower development (Fig. 10B).

In *S. densiflora*, the six stamens are arranged in the tepal radii, and initiate sequentially. The three stamens in the radii of the early-arising tepals initiate the first, and their primordia occupy the corners of the roundish-triangular floral apex inside the perianth (Fig. 10B). The other three stamens appear later between the first three stamens, and

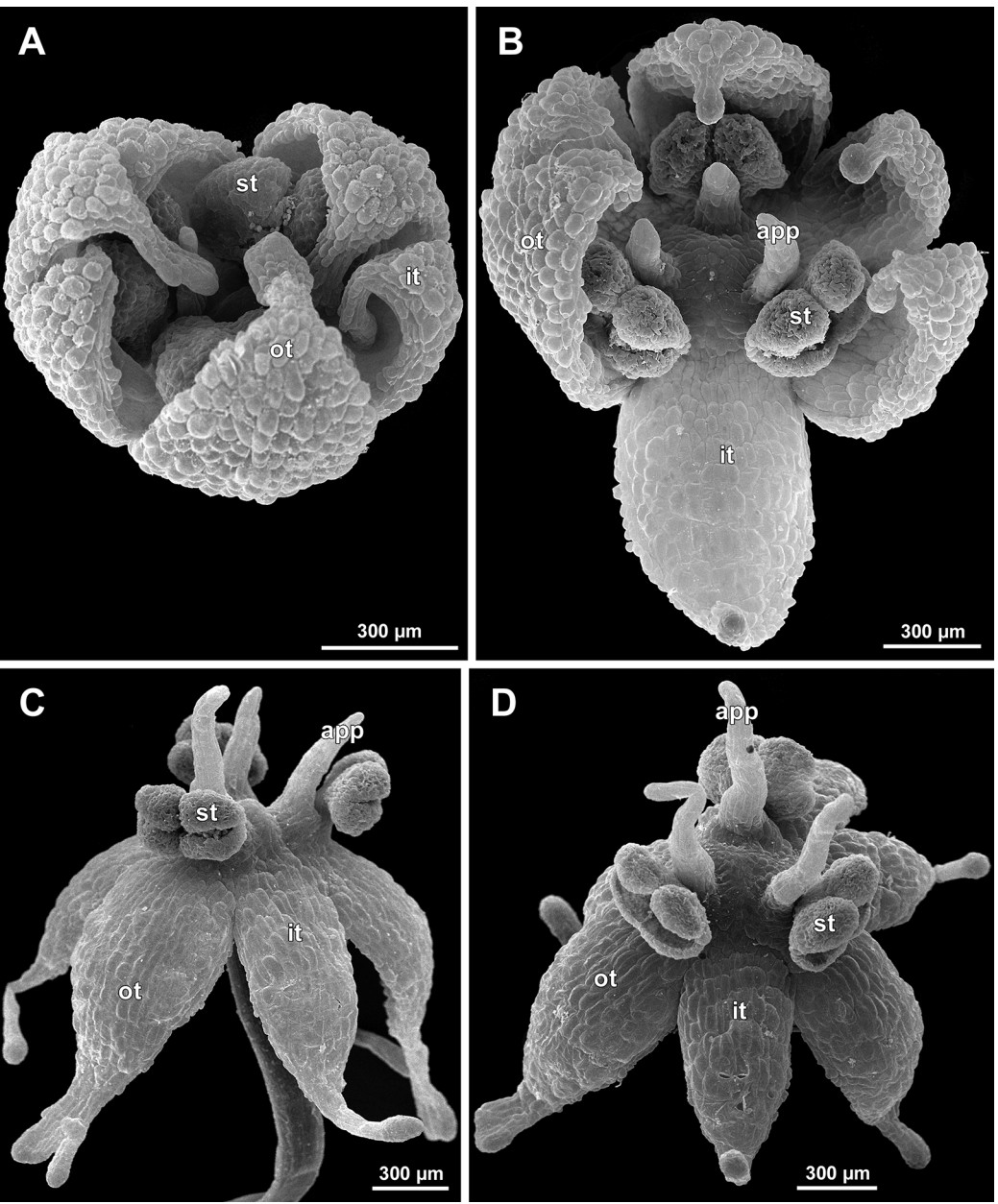

**Figure 9 *Sciaphila arfakiana*, male flowers (SEM) (*Nuraliev 1616*).** (A) Flower at late development, artificially opened, oblique view; note young stamen appendage. (B) Artificially opened preanthetic flower, oblique view. (C and D) Anthetic flowers, side and oblique view. Labels: app, stamen appendage; it, inner tepal; ot, outer tepal; st, stamen.               

during the early development of the androecium they remain smaller (Fig. 10D). In *S. arfakiana* (Fig. 7D) and *S. nana* (Fig. 14B), characterized by the androecium of three stamens, the stamen primordia occupy most of the space of the floral apex within the perianth, leaving a small unspent triangle of the floral apex between them. The stamens are arranged in the radii of early-arising tepals, that is, one stamen is median abaxial and two others are transversal-adaxial. In one of the flowers of *S. stellata* observed, only two

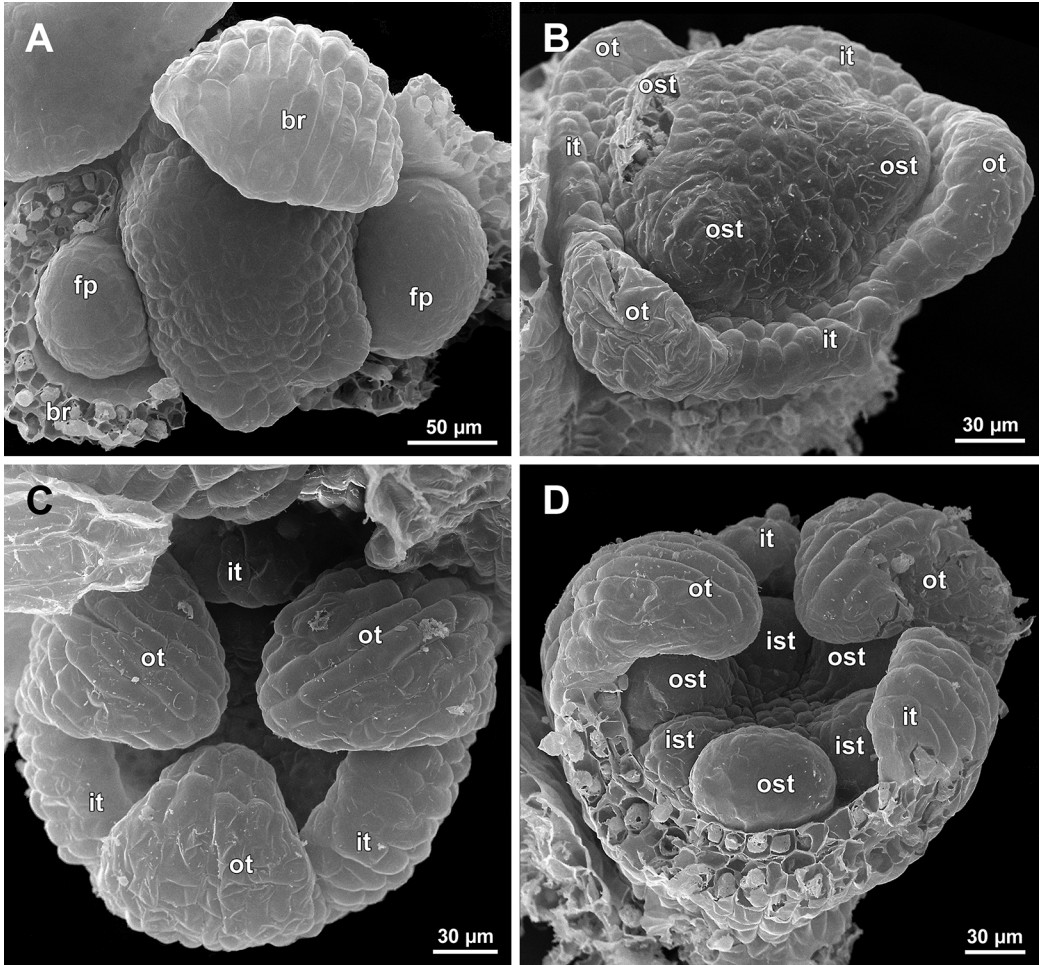

**Figure 10** *Sciaphila densiflora*, **early development of male flowers (SEM)** (*Nuraliev 1670*). (A) Distal part of inflorescence, top view. (B) Flower at initiation of free lobes of inner tepals and outer stamens, oblique view. (C) Flower at perianth development, top view with abaxial side facing down. (D) Flower at initiation of inner stamens, oblique view (one outer and one inner tepal removed). Labels: br, flower-subtending bract; fp, flower primordium; ist, inner stamen; it, inner tepal; ost, outer stamen; ot, outer tepal.

stamens were formed; additionally, one of the six tepals in this flower was much smaller than the others (Fig. 18B).

The stamens are initially hemispherical, but during their early development they intensively enlarge in the tangential direction (Figs. 8A, 11A and 14E). When a flower is ca. 250–300 μm in diameter, the stamens acquire a shape of a definitive anther, that is, they become about twice wider tangentially than radially (in top view) in *S. arfakiana* (Fig. 8B) and *S. nana* (Fig. 14F) and roundish-triangular in *S. densiflora* (Fig. 14E).

At this stage, the tepals become long and broad enough to entirely enclose the inner parts of the flower (Figs. 8C, 11A and 14F). Their free lobes undergo a much more intensive growth than the perianth tube, and the latter remains very short (and sometimes hardly recognizable) during further development. All six tepals acquire a triangular shape

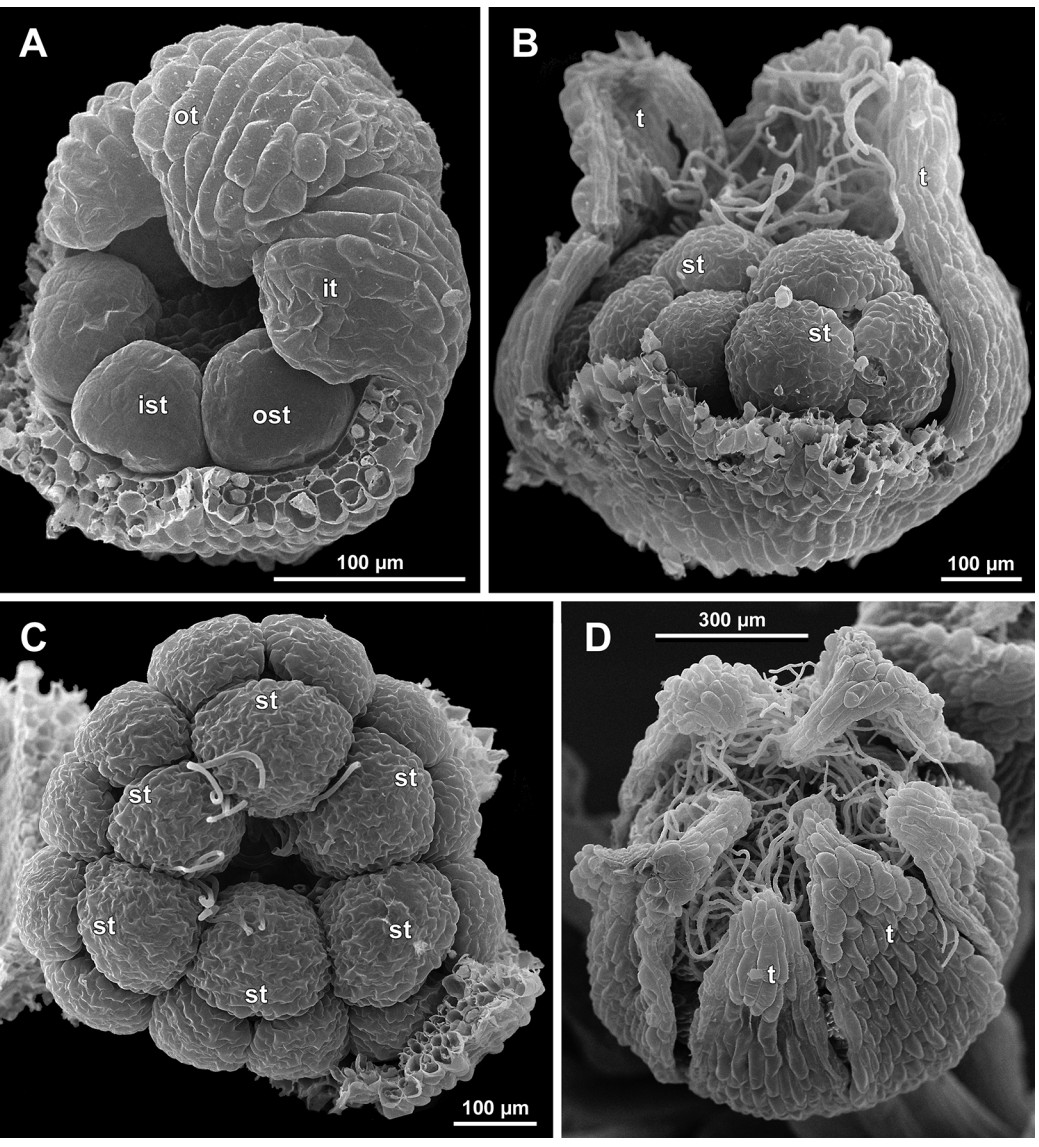

**Figure 11** *Sciaphila densiflora*, **development of male flowers (SEM)** (*Nuraliev 1670*). (A) Flower at initiation of anthers, oblique view (one inner and two outer tepals removed). (B) Preanthetic flower, side view (two tepals removed). (C) Preanthetic flower, top view (tepals removed). (D) Preanthetic flower, oblique view. Labels: ist, inner stamen; it, inner tepal; ost, outer stamen; ot, outer tepal; st, stamen; t, tepal.                

and a long attenuate apical part. They are appressed to each other with their lateral margins in a valvate manner. The tepals are curved inwards, and their apices point, and finally touch the undifferentiated floral center between the stamens. The attenuate distal parts of all six tepals thus contact each other in flower bud by their abaxial surfaces.

*Differentiation of tepals*

At late developmental stages, further differentiation of tepals and stamens takes place. The distal parts of tepals obtain their peculiar species-specific structure. In *S. stellata*, the tepals become pronouncedly caudate, that is, with a very long distal part, which is only

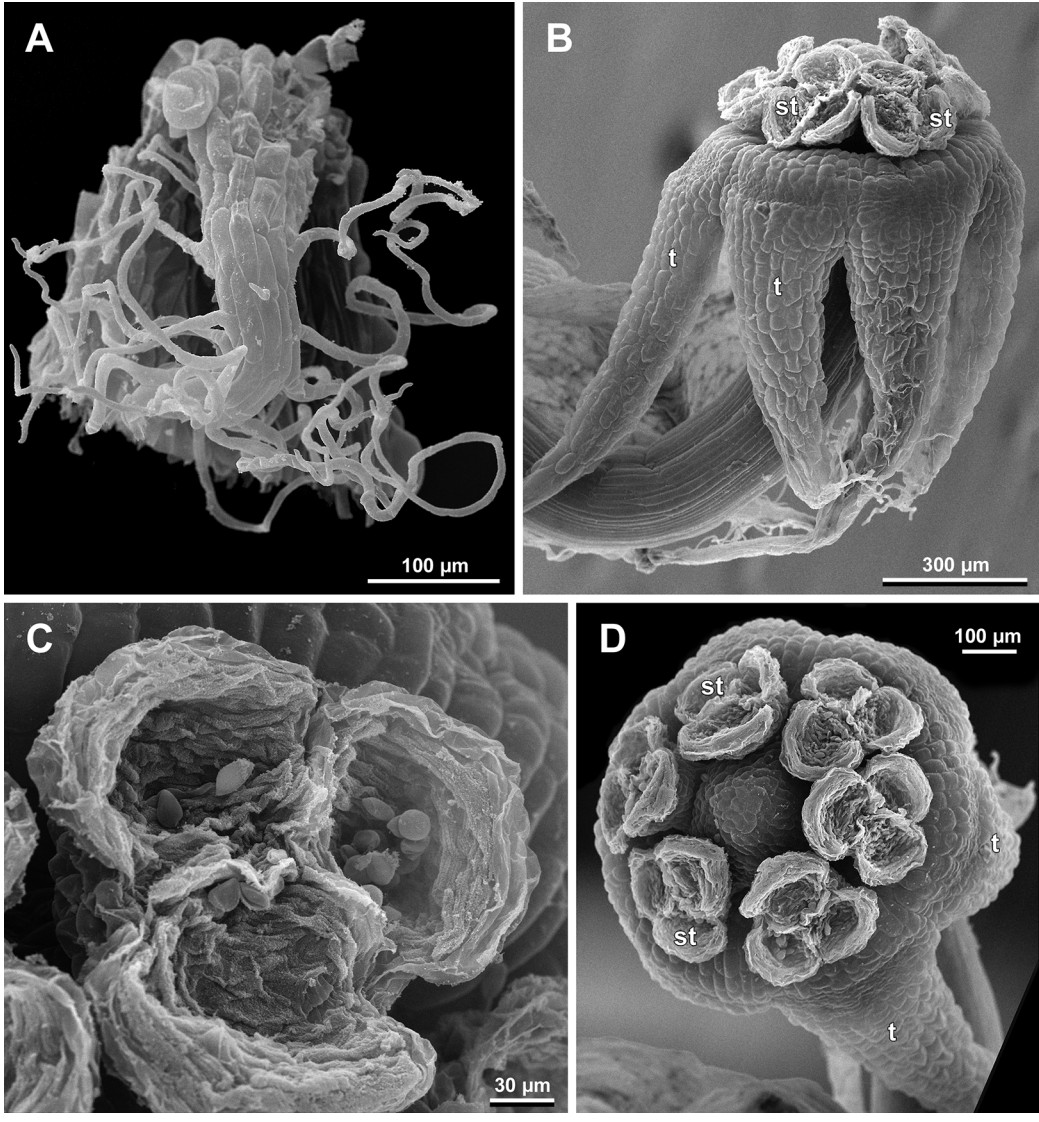

**Figure 12** *Sciaphila densiflora,* **male flowers (SEM)** (*Nuraliev 1670*). (A) Preanthetic free lobe of tepal, adaxial view, showing adaxially bent hairy apical portion. (B) Anthetic flower, side view. (C) Anther with dehisced locules, top view. (D) Anthetic flower, top view. Labels: st, stamen; t, tepal.

slightly shorter than the tepal blade (Figs. 17 and 18). In *S. densiflora*, the distal part of each tepal elongates significantly, and numerous long uniseriate multicellular hairs appear along the margin of the apical part (Figs. 11D, 12A and 12B). In *S. arfakiana*, all six tepals become distally not only attenuate but also thickened (clavate) at the very apex (Fig. 9). In *S. nana*, each of the three tepals alternating with stamens (i.e., the late-arising tepals) develops a distinctly attenuate and clavate distal part similar to that in *S. arfakiana*, whereas the tree tepals in the stamen radii (i.e., the early-arising tepals) do not obtain any thickenings, and during final stages become much less distinctly attenuate than at the middle developmental stages, ultimately being represented by just a blade with an acuminate apex (Figs. 15 and 16). The shape of the distal knob (assessed in preanthetic

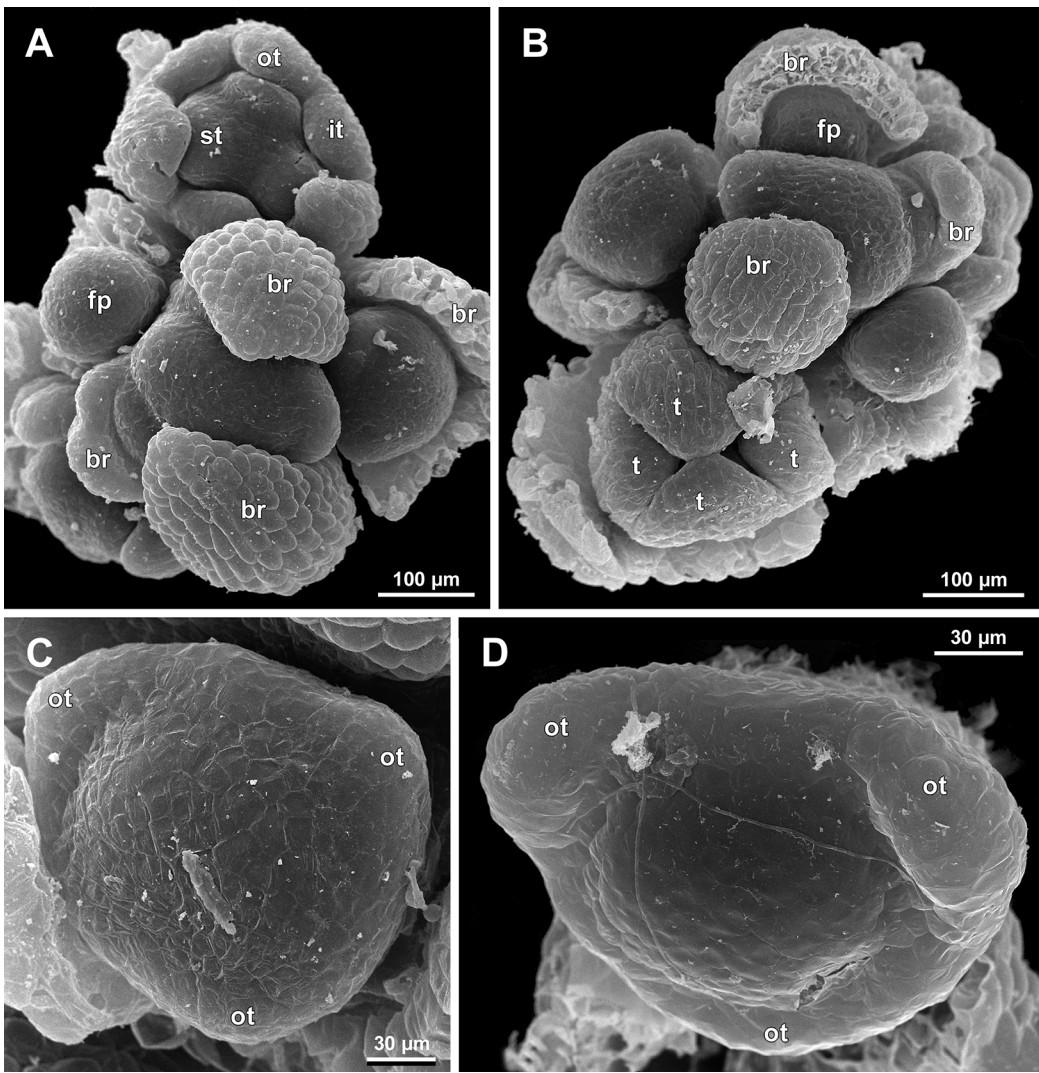

**Figure 13** *Sciaphila nana,* **early development of male flowers (SEM) (A–C:** *Nuraliev et al. 972,* **D:** *Nuraliev 1561).* (A and B) Distal part of inflorescence, top view; note a flower with four tepals in B. (C) Flower at initiation of outer tepals, top view with abaxial side facing down. (D) Flower with arising perianth tube, oblique view with abaxial side facing down. Labels: br, flower-subtending bract; fp, flower primordium; it, inner tepal; ot, outer tepal; st, stamen; t, tepal.

buds and open flowers) varies between the specimens of *S. nana* to a considerable extent: the knob is cylindrical (much longer than wide), with a gradual transition to the narrow portion of tepal in specimens *Nuraliev 498, Nuraliev 1669* (Fig. 15D) and *Nuraliev 2445* (Figs. 3B–3D and 16B); in contrast, the knob is nearly globose, sharply delimited from the narrow portion of tepal in *Nuraliev et al. 972* (Figs. 4B, 4C, 16A, 16C and 16D).

Distal portions of tepals are densely packed in late flower buds. They are placed between the stamens in the center of the floral bud, where the available space becomes strictly limited as the anthers grow. As observed in *S. arfakiana,* the distal portions of the

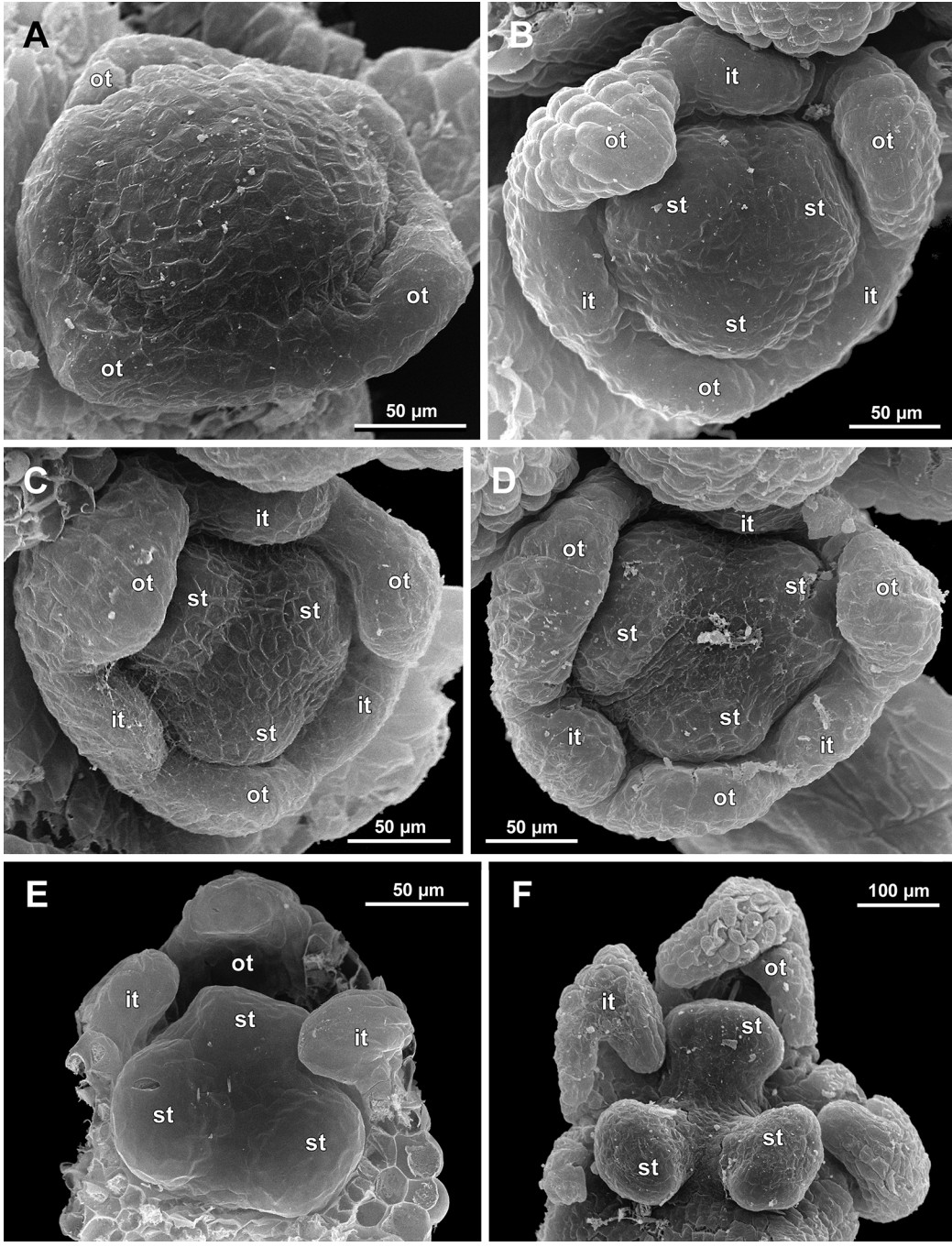

**Figure 14 *Sciaphila nana*, development of male flowers (SEM) (A–D, F: *Nuraliev et al. 972*, E: *Nuraliev & Kuznetsova 1380a*).** (A) Flower with first signs of free lobes of inner tepals, oblique view. (B–D) Flowers at subsequent stages of early development of stamens, top view with abaxial side facing down. (E) Flower with arising stamens, oblique view (two outer and one inner tepals removed). (F) Flower at initiation of anthers, oblique view (two outer and one inner tepals removed). Labels: it, inner tepal; ot, outer tepal; st, stamen.

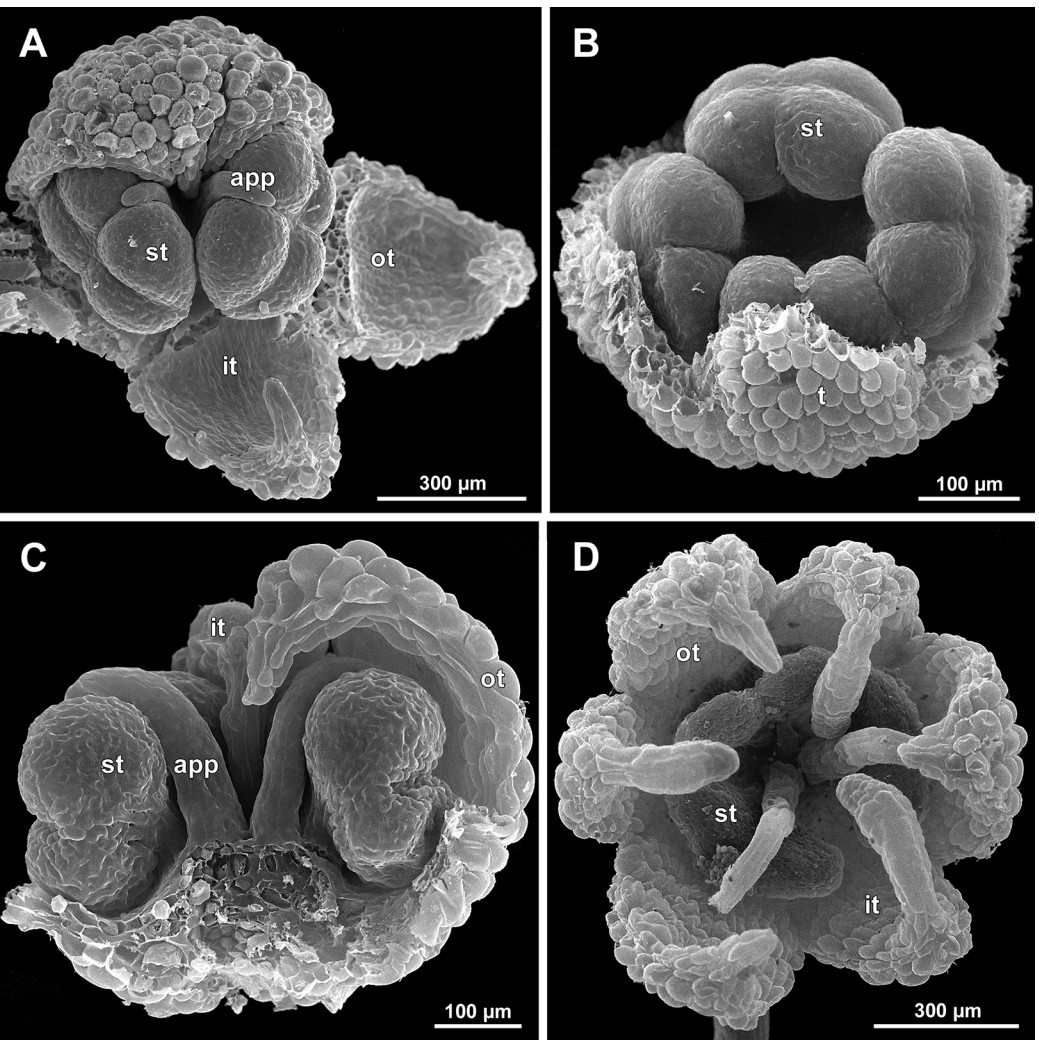

**Figure 15 *Sciaphila nana*, late development of male flowers (SEM) (A: *Nuraliev 498*, B: *Nuraliev 1561*, C: *Nuraliev 2445*, D: *Nuraliev 1669*).** (A) Artificially opened flower with differentiated anther locules, oblique view. (B) Flower with four stamens, top view (tepals removed). (C) Longitudinal half of flower, showing stamen appendages. (D) Artificially opened preanthetic flower. Labels: app, stamen appendage; it, inner tepal; ot, outer tepal; st, stamen; t, tepal.

antestaminous tepals are arranged exactly in the center, contacting each other, whereas those of the alternistaminous tepals are arranged more centrifugally, each of them constrained between two adjacent anthers and contacting two adjacent antestaminous tepals (Figs. 8C and 9A). For this reason, the antestaminous tepals possess more space to expand than the alternistaminous tepals. Consequently, the tepals of *S. arfakiana* are somewhat dimorphic with respect to their appendages in the preanthetic buds, with the appendages of antestaminous tepals being slightly wider (Fig. 9A), though the anthetic tepals are characterized by equal appendages in this species. The same pattern of arrangement of the tepal tips was documented in *S. nana* (Fig. 15A).

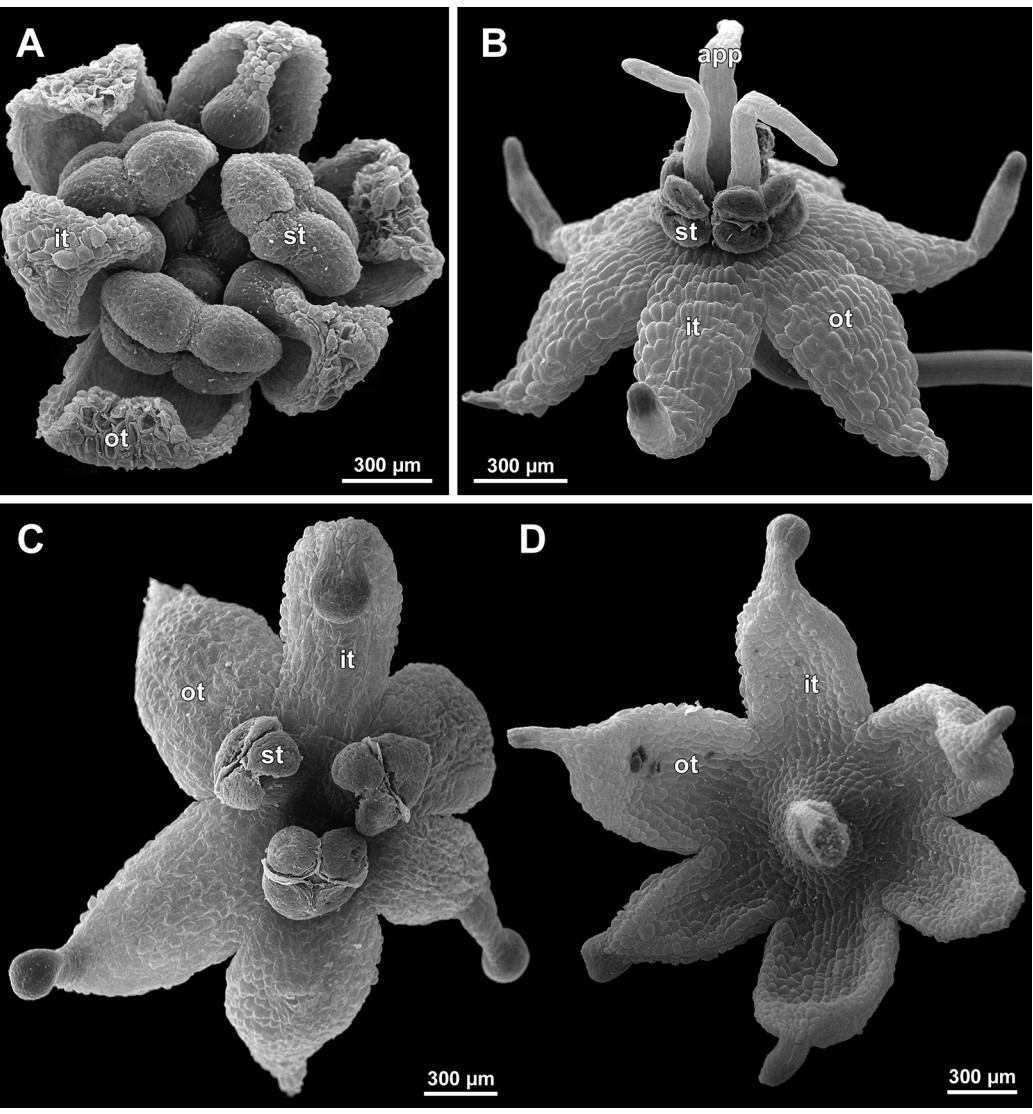

**Figure 16** *Sciaphila nana*, **male flowers (SEM) (A, C and D: *Nuraliev et al. 972*, B: *Nuraliev 2445*).**
(A) Artificially opened preanthetic flower. (B) Anthetic flower, oblique view; note long stamen appendages. (C) Anthetic flower, top view; note non-appendaged stamens. (D) Anthetic flower, view from below. Labels: app, stamen appendage; it, inner tepal; ot, outer tepal; st, stamen.

*Differentiation of stamens*

The stamens acquire their main structural parts during the late stages of flower development. First, the anther locules become discernible in each anther. In *S. densiflora*, three anther locules develop, a pair of which occupies an abaxial (outer) position, and a single (unpaired) one is in an adaxial (inner) position (Figs. 2A, 11B, 11C and 12B–12D). The unpaired locule is arranged in the symmetry plane of the stamen and is of more or less similar size with the paired ones (or just slightly larger), so that the three locules together form an isosceles triangle in the top view. All six stamens in a definitive flower of *S. densiflora* are uniform with respect to their size, structure and arrangement, and no

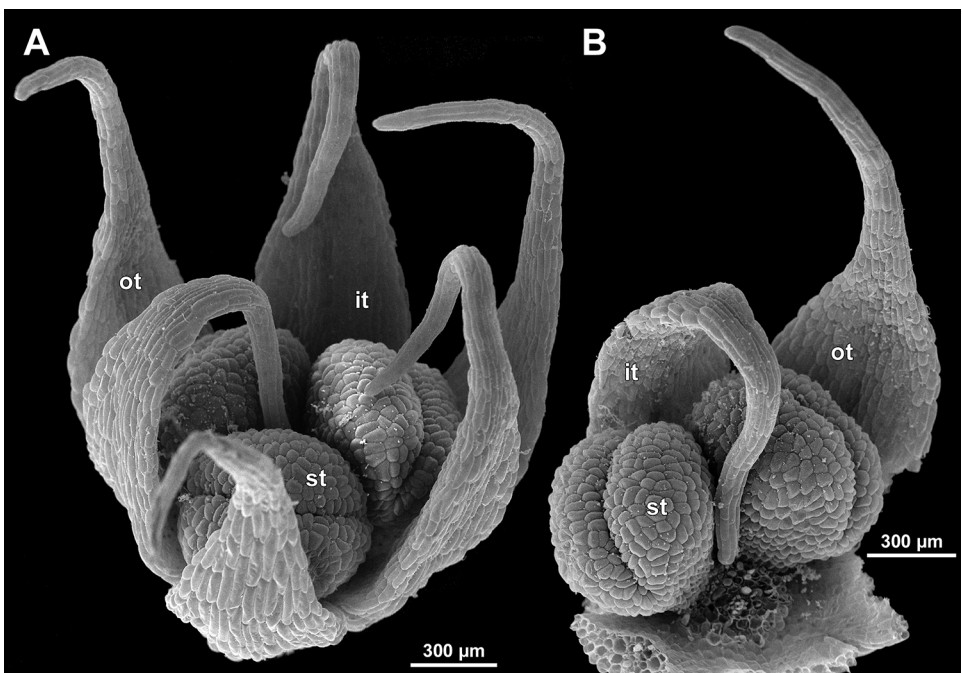

**Figure 17 _Sciaphila stellata,_ preanthetic male flowers (SEM) (_Nuraliev et al. 2499bis_).** (A) Oblique view. (B) Longitudinal half of the same flower as in A. Labels: it, inner tepal; ot, outer tepal; st, stamen.

direct evidence of their arrangement in two whorls can be found at this stage. In three other species (characterized by androecium of three stamens), the anthers are 4-locular, which is evident from the shape of both undehisced and dehisced anthers in _S. arfakiana_ (Figs. 1B, 8D and 9B–9D) and _S. nana_ (Figs. 15A, 15B, 16A and 16B). In _S. stellata_, the unopened and even the dehisced anthers have an appearance of 2-locular ones (Figs. 5B, 17, 18A and 18B), because only the transversal boundary between the microsporangia is visible, but not the median boundary (delimiting left and right thecae). However, the presence of four locules is clearly seen in a sectioned anther of _S. stellata_ (Fig. 18C). Additionally, an apparently 3-locular condition was observed in undehisced anthers of certain flowers of _S. nana_: the median boundary between the thecae in the abaxial half of the anther was hardly recognizable, in contrast to the adaxial half (Fig. 16C). However, the presence of truly 3-locular anthers in this species was not confirmed by observations of dehisced anthers.

Tetrasporangiate anthers dehisce by a long transversal slit. In _S. arfakiana_ (Fig. 9D) and _S. nana_, remnants of the four septae between the anther locules are often recognizable in dehisced anthers, but these are no longer visible in _S. stellata_ (Figs. 18A and 18B). Trilocular anthers of _S. densiflora_ dehisce by three large slits: each locule opens by an individual transversal slit facing towards the anther centre (Figs. 12B–12D).

During the anther differentiation, a filament develops in each stamen as a result of intercalary growth between the anther and the stamen base. The filament is usually much shorter than the anther and evidently to only slightly narrower in diameter than the anther (Figs. 9C, 12D, 15C and 17B).

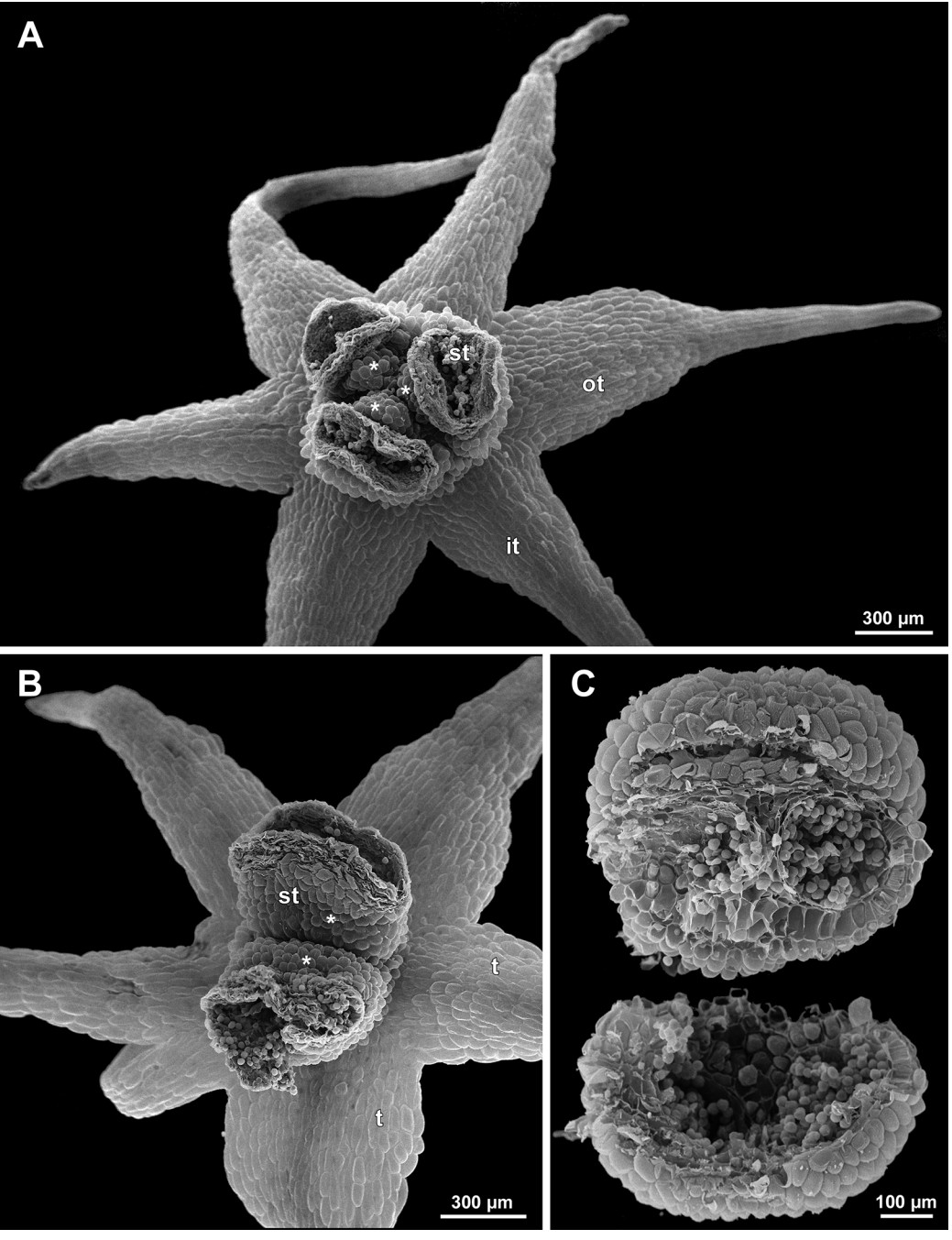

**Figure 18** *Sciaphila stellata*, **male flowers (SEM) (A and B:** *Nuraliev & Kuznetsova 1380b*, **C:** *Nuraliev et al. 2499bis*)**.** (A) Anthetic flower, top view. (B) Anthetic flower with two stamens, oblique view. (C) Anther of the same flower as in Fig. 17 with one transversal half artificially opened; note two locules in the opened part. Labels: it, inner tepal; ot, outer tepal; st, stamen; t, tepal. Asterisk indicates a stamen filament.

Stamen appendages appear at very late stages of flower development. The presence of the appendages is variable among and in some cases within the species under study, but seems to be constant within the populations. The appendages were observed in both

specimens of *S. arfakiana* (Figs. 1A, 1B and 9), and their complete absence was documented for the specimens of *S. densiflora* (Figs. 2 and 12D) and *S. stellata* (Figs. 5B, 17B, 18A and 18B). Among the specimens of *S. nana*, prominent appendages were observed in *Nuraliev 498* (Fig. 15A), *Nuraliev & Kuznetsova 1380a*, *Nuraliev 1669*, *Nuraliev 2445* (Figs. 3A–3D, 15C and 16B) and they were absent in *Nuraliev et al. 972* (Figs. 4, 16A and 16C) and possibly in *Nuraliev 1561* (the data obtained for the latter specimen are inconclusive; Fig. 15B). In anthetic flowers, the stamen appendages usually greatly (sometimes several times) exceed the anthers in length, being comparable in their size with the tepals. The anthers are extrorse in flowers with stamen appendages and nearly latrorse with almost vertically oriented dehiscence lines in flowers without the appendages. The appendages constantly occupy a precise adaxial position with respect to the (rest of the) stamen. The process of the appendage initiation has not been observed. The evident but small appendages (much shorter than anthers) were recorded in almost completely developed flowers of about 600–700 μm in diameter (Fig. 9A). In preanthetic buds, the appendages are strongly bent away from the flower center and tightly appressed to the anther, lying in the depression between the thecae (Figs. 15A and 15C). In *S. arfakiana*, a stamen and an appendage in its radius possess a prominent common base (Figs. 9C and 9D), which cannot be interpreted with confidence either as a stamen structure or an outgrowth of the receptacle. In *S. nana*, in contrast, the appendage is almost free from the stamen, without any prominent common tissue (Fig. 15C).

The floral center between the stamens (and their appendages, if present) is slightly elevated and flat in *S. arfakiana* (Fig. 9) and *S. nana* (Figs. 16A and 16C), and forms a prominent conical apex in *S. densiflora* (Figs. 12B and 12D). An apparent exception here is *S. stellata*, whose thick, sometimes almost cushion-shaped stamen filaments are tightly arranged in anthetic flowers, occupying the floral center (Figs. 17B, 18A and 18B).

**Female flowers (Figs. 19–30)**

*Flower at initiation and development of gynoecium*
Floral development was traced beginning from the stage of initiation of the first carpels, when the flower bud is ca. 300–350 μm in diameter (exemplified by *S. arfakiana*, Fig. 19A) and possesses a perianth almost entirely covering the gynoecium. At this stage, the floral apex inside the perianth is circular in the outline and has a shape of a shallow dome. The carpels are initiated in an acropetal sequence. The first-formed carpel primordia are arranged in a slightly irregular whorl (Figs. 19A and 19B). Subsequent carpels appear in a staggered order, more or less alternating with the previously initiated carpels (Figs. 19C, 19D, 20, 21A, 21B and 26). This process continues until the floral meristem is completely exhausted, and no residual floral apex is visible after all the carpels are initiated (Figs. 21C, 21D, 22A, 26F, 27, 28B–28D and 29B–29D). In a developing gynoecium, more proximal carpels are significantly larger and more differentiated than the more distal ones.

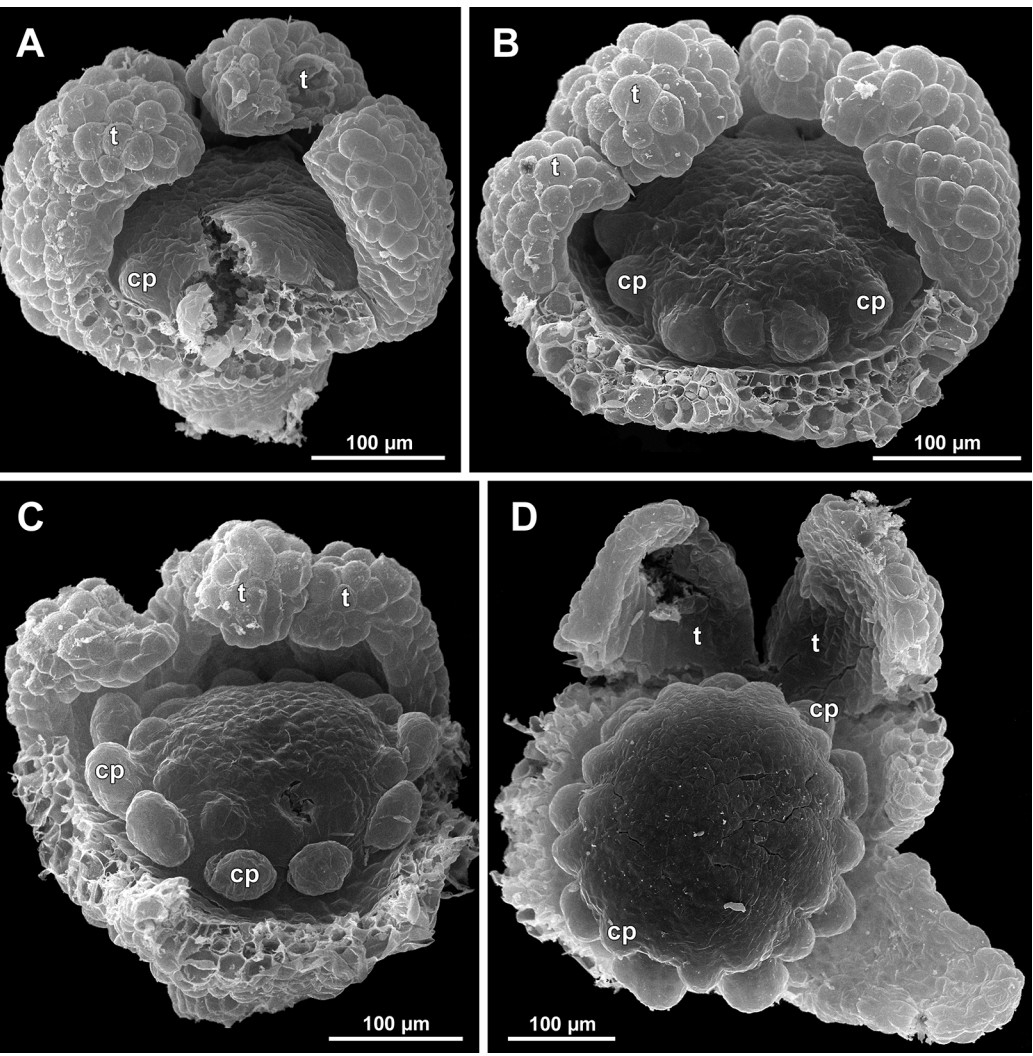

**Figure 19 *Sciaphila arfakiana*, early development of female flowers (SEM) (*Nuraliev 1616*).**
(A) Flower at inception of first carpels, side view (three tepals removed). (B and C) Flowers at early gynoecium development, side view (some tepals removed). (D) Flower at early gynoecium development, top view (three tepals removed). Labels: cp, carpel primordium; t, tepal.

The female flowers of *S. arfakiana*, *S. nana* and *S. stellata* were proved to be morphologically unisexual (lacking any androecial structures), whereas in *S. densiflora* we observed organs which we interpret as staminodes. We investigated staminodes in detail in two flowers bearing six (Fig. 24A) and seven (Fig. 24D) of them. The staminodes are attached to the receptacle between the tepals and the basalmost carpels, tending to occupy the tepal radii: at least, the arrangement of the staminodes in tepal radii was found in a flower with six staminodes (Figs. 23, 24A and 24B). The anthetic staminodes are much smaller than the carpels, ca. 100 μm long and wide, and possess an irregular shape ranging from an elliptic to a rectangular one (Figs. 24A, 24B and 24D). They are constrained between the perianth tube and the carpels and thus are completely invisible in

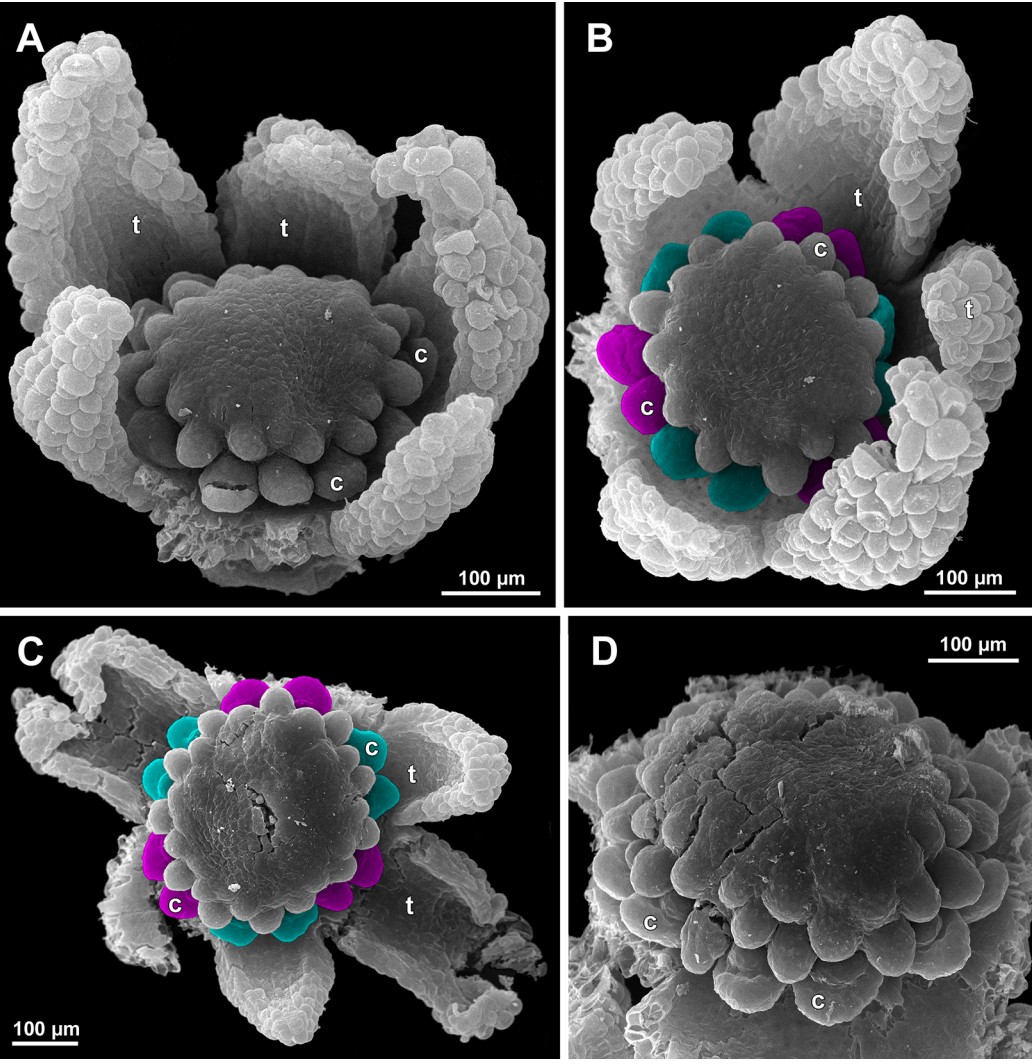

**Figure 20** *Sciaphila arfakiana*, **development of female flowers (SEM)** (*Nuraliev 1616*). (A) Flower at early gynoecium development, oblique view (one tepal removed). (B and C) Flowers at early gynoecium development, top view (some tepals removed); pairs of the outermost carpels are coloured according to their arrangement in tepal sectors. (D) Gynoecium with asciidiate zone visible in the outermost carpels, oblique view. Labels: c, carpel; t, tepal.

the intact flower (Fig. 23); it is necessary to remove the gynoecium from the flower to uncover them.

*Development of individual carpels*

A carpel initiates in the form of a hemispherical primordium 20–30 μm in diameter (Figs. 19–21, 26 and 28B–28D). It begins to elongate, and when the carpel becomes ca. 50 μm in diameter and about twice as long as wide, it differentiates into a scoop-like blade (carpel wall) at dorsal side and a short, only slightly convex outgrowth at ventral side. The ventral outgrowth is divided from the dorsal blade by a shallow depression, and there are low flanges connecting the dorsal and ventral parts at left and right margins of the depression (Figs. 20D, 21A, 21B, 26C, 26D and 28B–28D). The ventral part soon gives rise

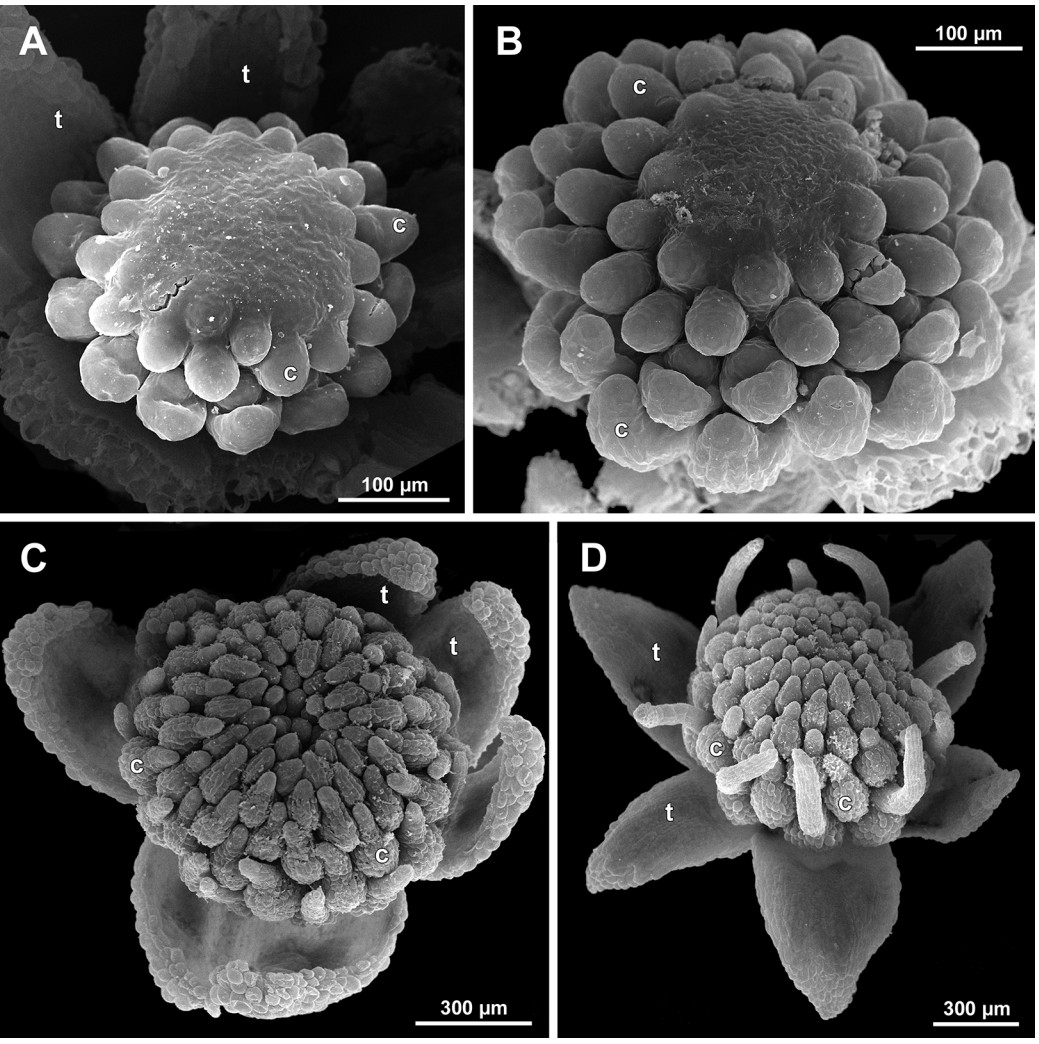

**Figure 21** *Sciaphila arfakiana*, **development of female flowers (SEM) (A–C:** *Nuraliev 1616*, **D:** *Nuraliev et al. 2499*). (A and B) Gynoecia with the innermost carpels not yet initiated, oblique view. (C) Flower with all carpels initiated, top view. (D) Flower at elongation of styles of the outermost carpels, oblique view. Labels: c, carpel; t, tepal.

to the ovule; it is not possible to determine the exact stage of ovule initiation, because the ovule is formed from almost the entire ventral part. The ovule is thus inserted close to the carpel base, at the ventral side of the carpel (rather than at the dorsal blade or basally). Usually, there is a small portion of carpellary tissue separating the ovule from the receptacle. After the ovule inception, the ventral carpel wall ceases its growth and remains short during subsequent stages. During further growth, the dorsal part of the carpel wall curves towards the floral center and covers the ovule in a hood-like manner. When the carpel is more than 100 μm wide, its blade encloses the ovule with blade margins. A ventral slit becomes sealed through postgenital fusion, completing the formation of an ovary locule. The apex of the blade becomes attenuate and differentiates into a style which undergoes extensive elongation at further stages of development. The style is entirely cylindrical and lacks a ventral furrow (Figs. 22, 25D, 29 and 30). Simultaneously with the

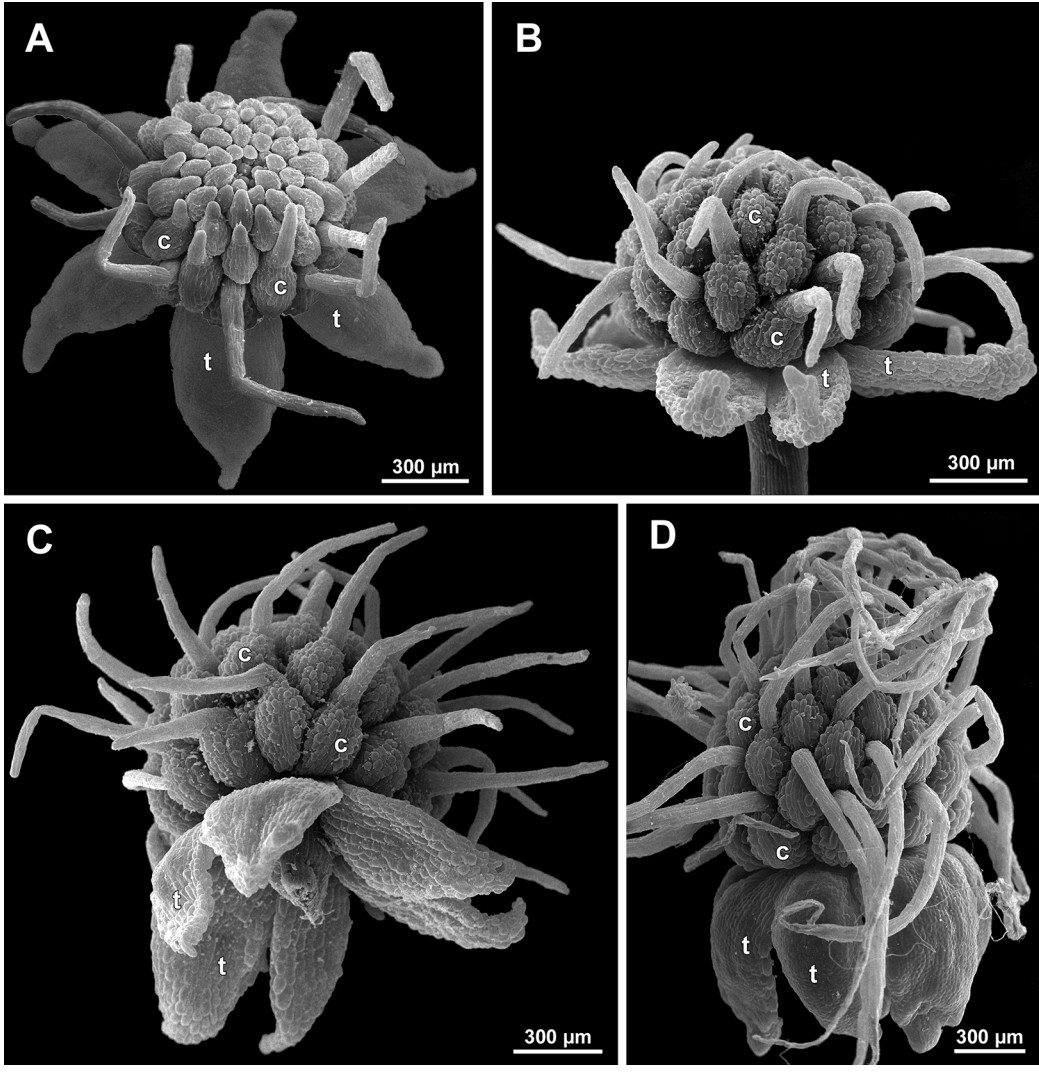

**Figure 22 *Sciaphila arfakiana*, anthetic female flowers (SEM) (A and D: *Nuraliev et al. 2499*, B and C: *Nuraliev 1616*).** (A) Flower at early anthesis, with distal carpels still underdeveloped, oblique view. (B and C) Flowers at mid-anthesis, side view. (D) Flower at late anthesis, side view. Labels: c, carpel; t, tepal.

style elongation, the locule expands at its dorsal side, so that the style appears to be a continuation of the ventral rib of the ovary (bearing the ventral slit) during all subsequent developmental stages. At late stages, the dorsal expansion of the locule is much faster than its growth in the ventral region; eventually, the ventral rib becomes significantly shorter than the ovary, and accordingly the style acquires a gynobasic position. In anthetic carpels, the ventral rib, and therefore the ventral slit, is vanishingly short and hardly recognizable.

*Phyllotaxis of gynoecium*
In *S. densiflora* (characterized by morphologically bisexual flowers), there is a precise alternation of the staminodes and the outermost carpels (Figs. 24A, 24B and 24D).

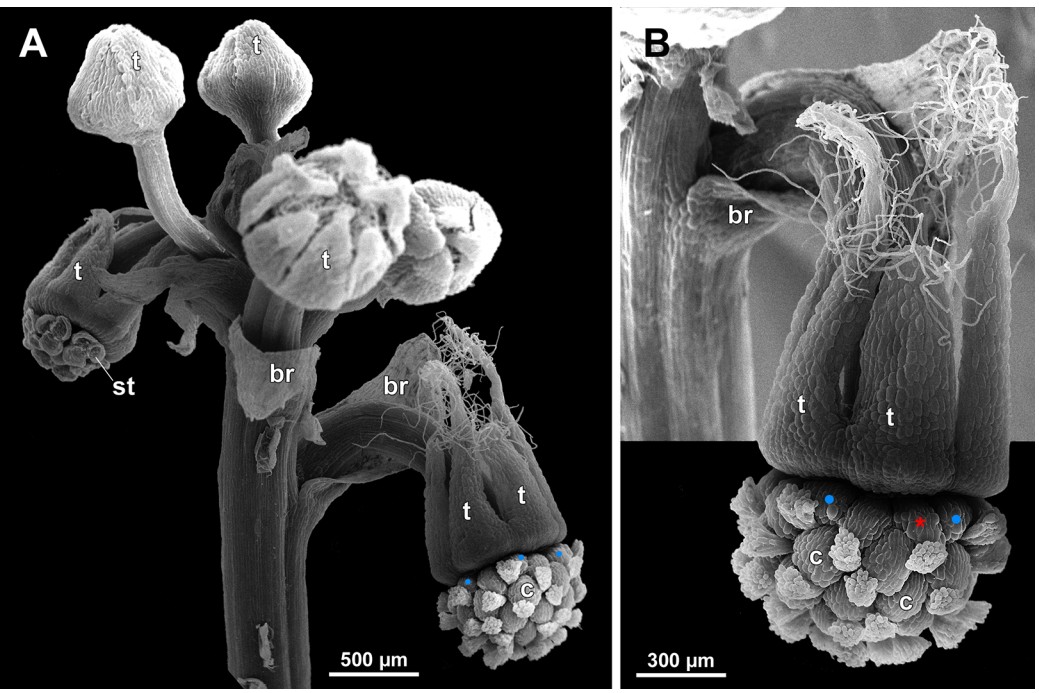

**Figure 23** *Sciaphila densiflora* (SEM) (*Nuraliev 1670*). (A) Part of inflorescence, showing five male and one female flowers, side view. (B) Female flower, side view. Labels: br, flower-subtending bract; c, carpel; st, stamen; t, tepal. Blue circles indicate carpels of the outermost whorl (alternating with staminodes); red asterisk indicates an individual carpel which is also visible in Fig. 24B and allows to match the images in order to establish the position of the staminode with respect to the tepals.

Thus, there are six or seven outermost carpels, and they tend to alternate with the tepals (Figs. 23 and 24).

In the studied species with morphologically unisexual flowers, there are usually about 12 carpels in the outermost set (i.e., twice as many as the tepals). These carpels apparently form a whorl. No uniform regularity of arrangement of these carpels with respect to the tepals was detected. Analysis of this character is technically difficult, because one needs to dissect and partly remove the perianth for observation of the youngest carpels, so that the information of tepal arrangement is often being lost. The most appropriate material for investigation of the disposition of the outermost carpels was obtained for *S. arfakiana*. Two studied flowers of this species possess two carpels in sector of each tepal, with more or less similar angular distance within and between the carpel pairs (Figs. 20B and 20C). Apart from the common situation of the presence of twelve outermost carpels, we observed one flower with eleven and another flower with nine outermost carpels. In the case of eleven carpels (Fig. 19C), at least two carpels are inserted as alternating with adjacent tepals, and the position of the others is less obvious. The occurrence of some outermost carpels in radii between adjacent tepals is also documented in several flowers of *S. arfakiana* for which we failed to document arrangement of every floral organ, because some sectors were damaged during dissection (Fig. 19A, 19B, 20D and 21A). In a flower of *S. arfakiana* with nine outermost carpels

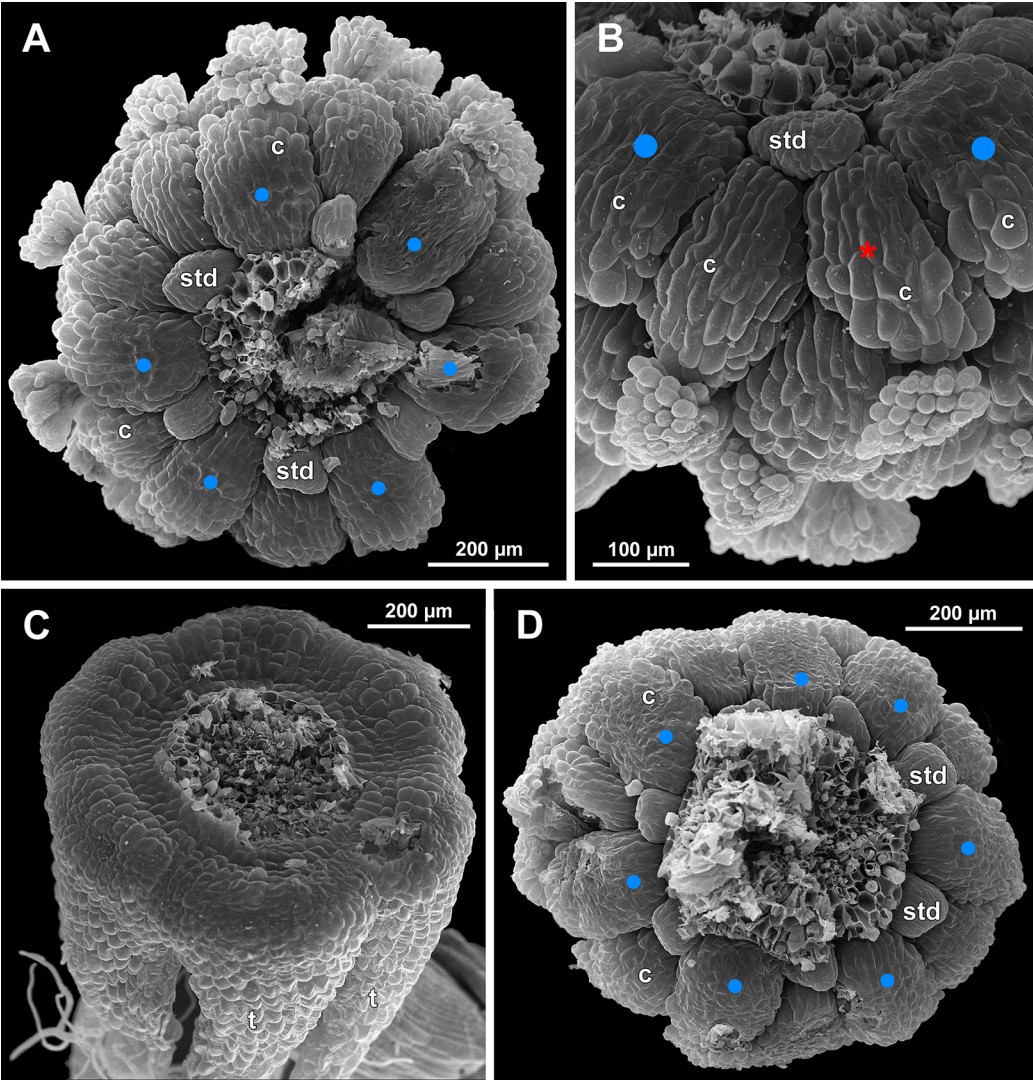

**Figure 24** *Sciaphila densiflora*, **anthetic female flowers (SEM) (*Nuraliev 1670*).** (A) Gynoecium detached from perianth, view from below, showing six staminodes. (B) Portion of the same flower as in A, view from below and outside, showing arrangement of staminodes and carpels. (C) The same flower as in A, showing perianth after removal of gynoecium, oblique view. (D) Gynoecium removed from perianth, view from below, showing seven staminodes. Labels: c, carpel; std, staminode; t, tepal. Blue circles indicate carpels of the outermost whorl (alternating with staminodes); red asterisk indicates an individual carpel which is also visible in Fig. 23B and allows to match the images in order to establish the position of the staminode with respect to the tepals.

(Fig. 19D), information on position of three adjacent tepals is available. The central tepal of this group (which is wider than the two others) has two outermost carpels in its sector. Of the two other visible tepals, one has two outermost carpels in its sector and the other has only one carpel in its sector while there is an outermost carpel between this and an adjacent removed tepal. The space left after the removed tepals (much less than 1/2 of flower perimeter) suggests that this flower with nine outermost carpels had five rather than six tepals. In a flower of *S. nana* with six tepals, nine outermost carpels were also detected,

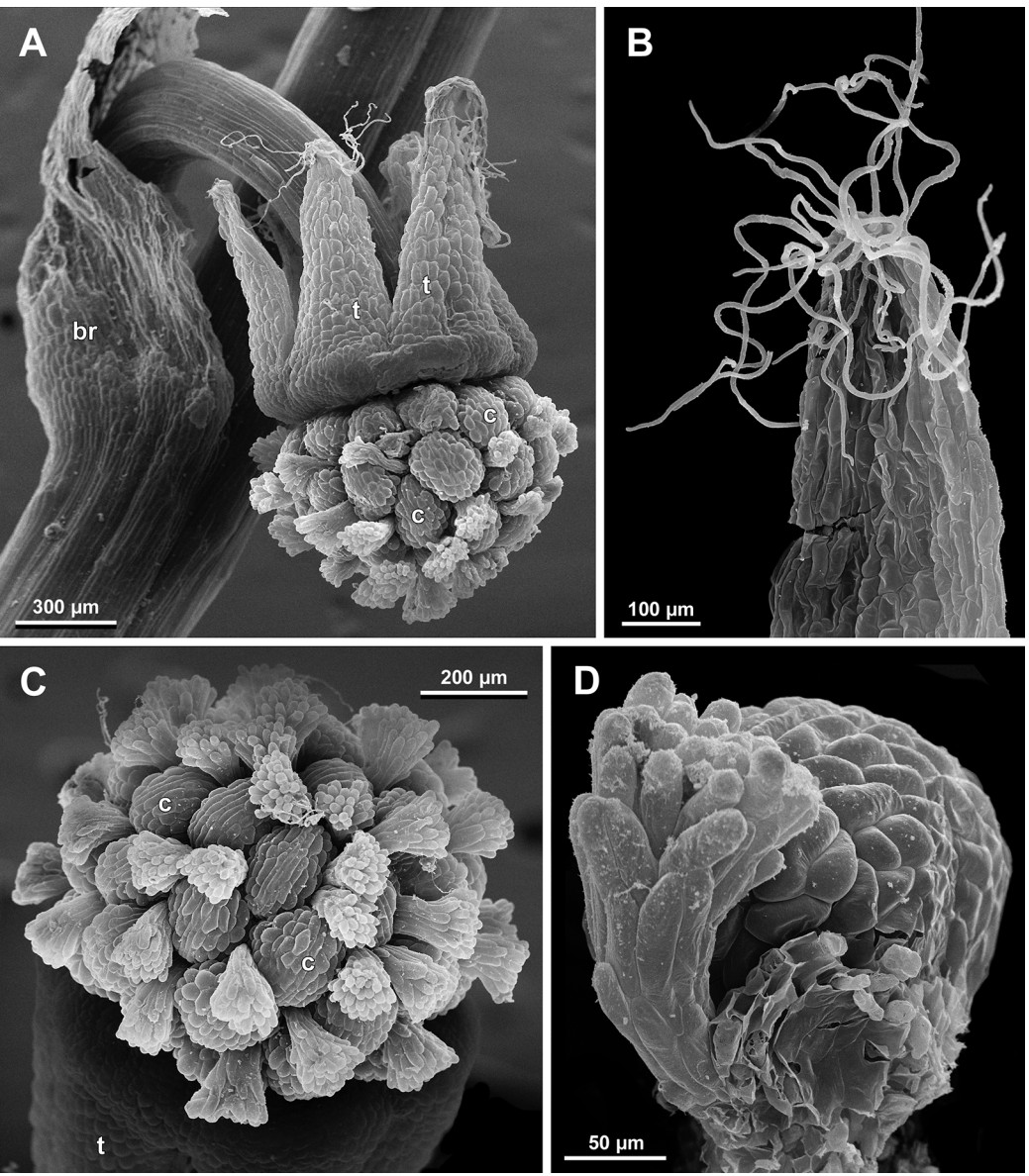

**Figure 25** *Sciaphila densiflora*, **female flowers (SEM)** (*Nuraliev 1670*). (A) Anthetic flower, side view. (B) Apical part of preanthetic tepal, adaxial view. (C) Anthetic gynoecium, oblique view. (D) Carpel, side view. Labels: br, flower-subtending bract; c, carpel; t, tepal.

but their arrangement was highly irregular (Fig. 26A). Indeed, the angular distance between adjacent outermost carpels varied from 24° to 65°.

We investigated details of carpel arrangement in entire gynoecia of three flowers of *S. nana* using their SEM images. These flowers possessed all their carpels developed or at least initiated, and at the same time they were young enough to show clearly the position of each carpel on the receptacle. The images chosen for this analysis are top views of gynoecium with all the carpels being visible. We attempted to recognize contact parastichies in the gynoecium, one set of them in each direction. In uncertain cases,

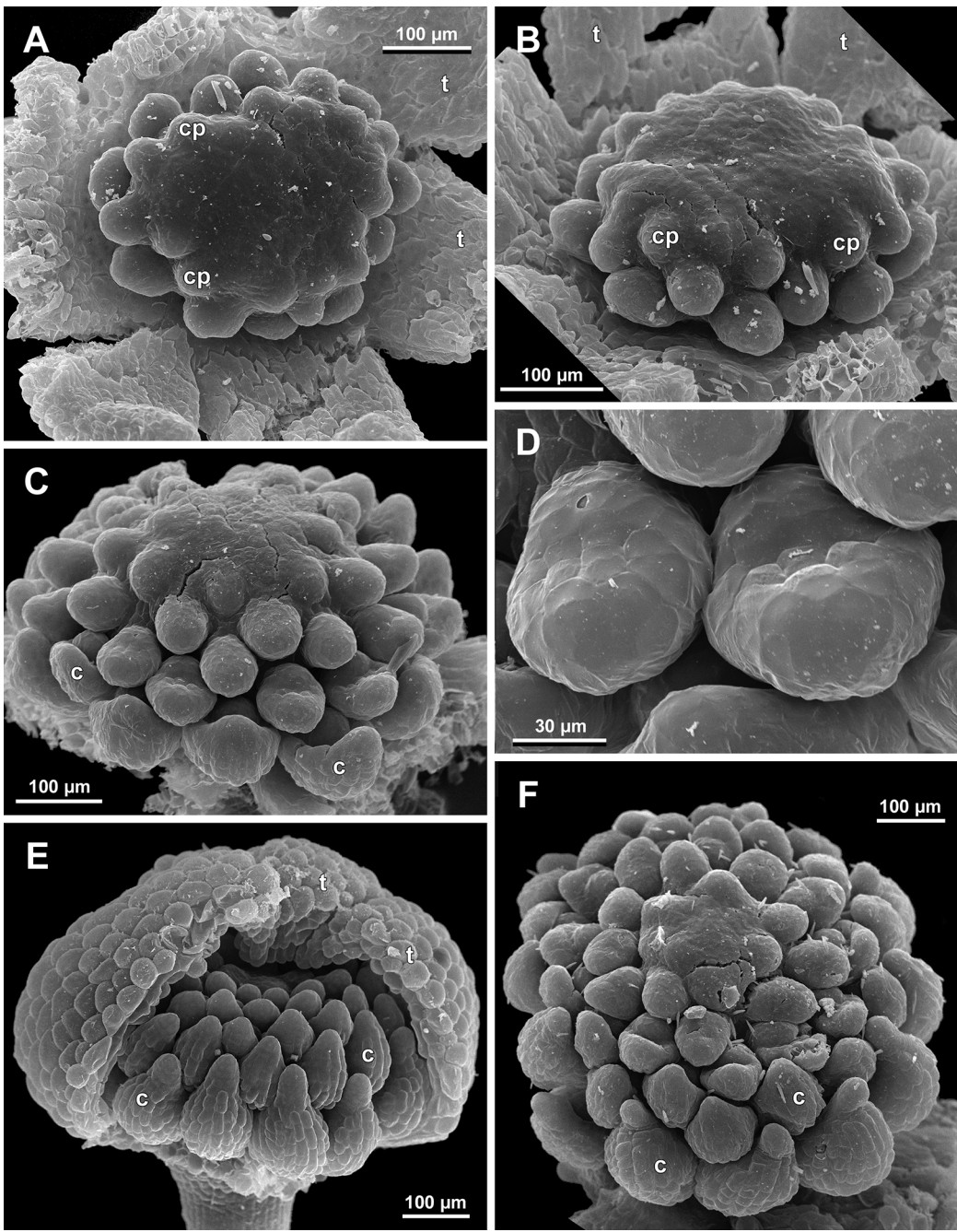

**Figure 26** *Sciaphila nana*, development of female flowers (SEM) (A–D, F: *Nuraliev et al. 972*, E: *Nuraliev 1561*). (A and B) Gynoecia at early development, top and oblique view. (C) Gynoecium at initiation of inner carpels, oblique view. (D) Carpels prior to ovule initiation, top view, showing ascidiate zones. (E) Flower at initiation of the innermost carpels, side view (some tepals removed). (F) Gynoecium at initiation of the innermost carpels, top view. Labels: c, carpel; cp, carpel primordium; t, tepal.

we chose options that allowed to minimize the number of carpels not included into any parastichy. We also tended to follow the idea that the parastichies within a given gynoecium should contain similar carpel numbers. In all three flowers, we found it

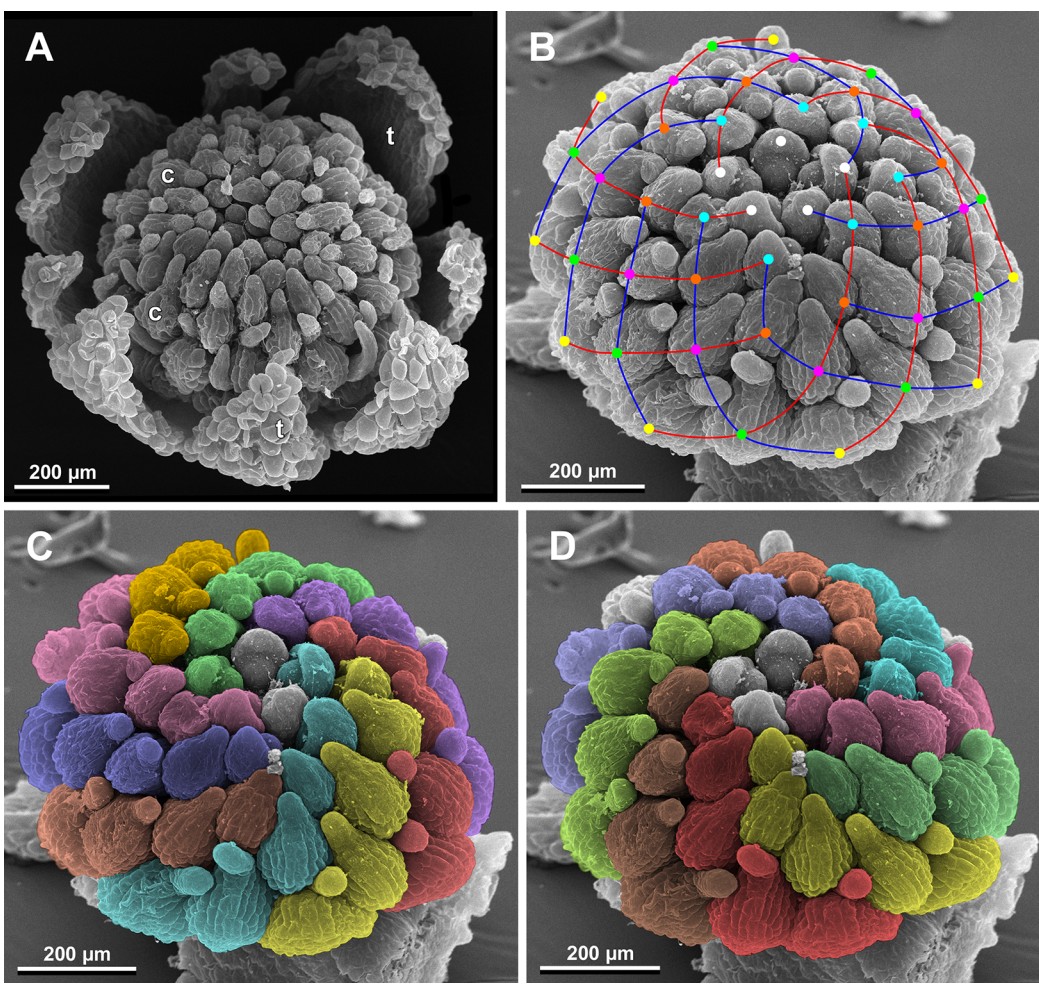

**Figure 27** *Sciaphila nana*, **late development of female flowers (SEM) (A, *Nuraliev & Kuznetsova 1380a*, B–D: *Nuraliev et al. 972*).** (A) Flower at elongation of styles of the outermost carpels, top view. (B) Gynoecium, oblique view, with superimposed parastichies of opposite directions indicated by red and blue lines; colour circles indicate carpel whorls. (C, D) The same image as in B with carpels coloured according to estimated parastichies. Labels: c, carpel; t, tepal.

impossible to attribute the innermost carpels to any of the parastichies. In one of the flowers (Figs. 27B–27D), we were able to trace quite smooth parastichies, with only one of the basalmost carpels left unmarked. There are nine parastichies in each direction. The parastichies contain from four to six, most commonly five carpels. On the basis of the parastichies, six whorls of the following merism are recognized (in acropetal sequence): 9 (including the unmarked carpel), 9, 9, 9, 7, 5. In the second flower (Figs. 28B–28D), the recognition of the parastichies was less straightforward. Although both right and left parastichies inferred from our analysis comprise most of the carpels, the parastichies show rather uneven angles of curvature and significantly differ from each other in the number of carpels: there are from three to six carpels per a parastichy, with the cases of four, five and six carpels being approximately equally common. There are 10 right and

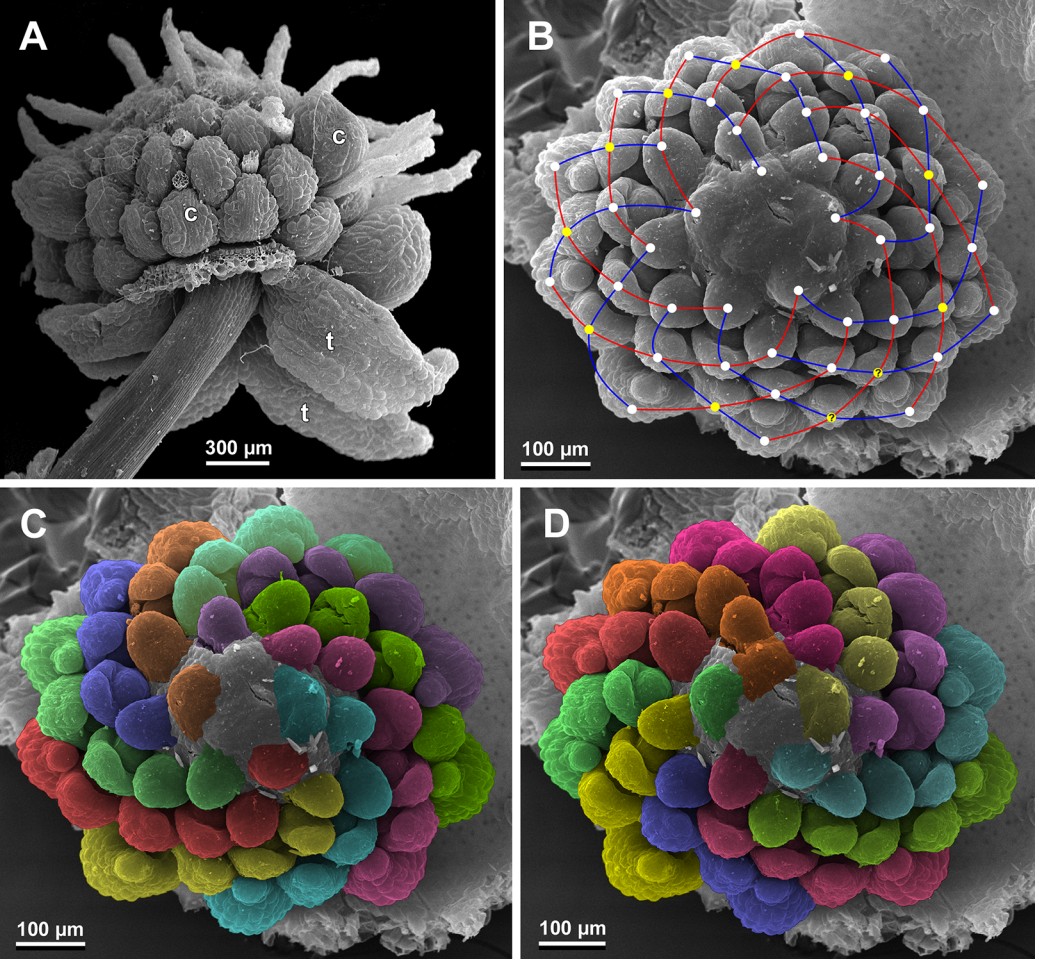

**Figure 28** *Sciaphila nana*, **development of female flowers (SEM)** (*Nuraliev et al. 972*). (A) Anthetic flower, side view. (B) Gynoecium, top view, with superimposed parastichies of opposite directions indicated by red and blue lines; yellow circles indicate an attempt to recognize a carpel whorl; this whorl seems to continue into the next whorl, which is consistent with the idea of non-integer merism. (C and D) The same image as in B with carpels coloured according to estimated parastichies. Labels: c, carpel; t, tepal.

11 left parastichies. In accordance with the unequal number of right and left parastichies, no whorls were detected in this gynoecium. In the third flower (Figs. 29B–29D), the parastichies could hardly be detected. Our best attempt resulted in a remarkably irregular diagram, in which six carpels belong each to a single parastichy (instead of two parastichies of opposite directions), three parastichies contain three carpels each (whereas the others have 4–6 carpels), and neighboring parastichies of the same direction frequently meet at almost a straight angle. Within the illustrated interpretation, there are eight parastichies in each direction, and five whorls of carpels characterized by the following merism: 10, 8, 8, 5, 5. In this flower, one of the carpels possessed an apparently aberrant orientation, as if it has been turned at a right angle from its normal position; its style was pointing the neighboring carpel instead of the floral apex.

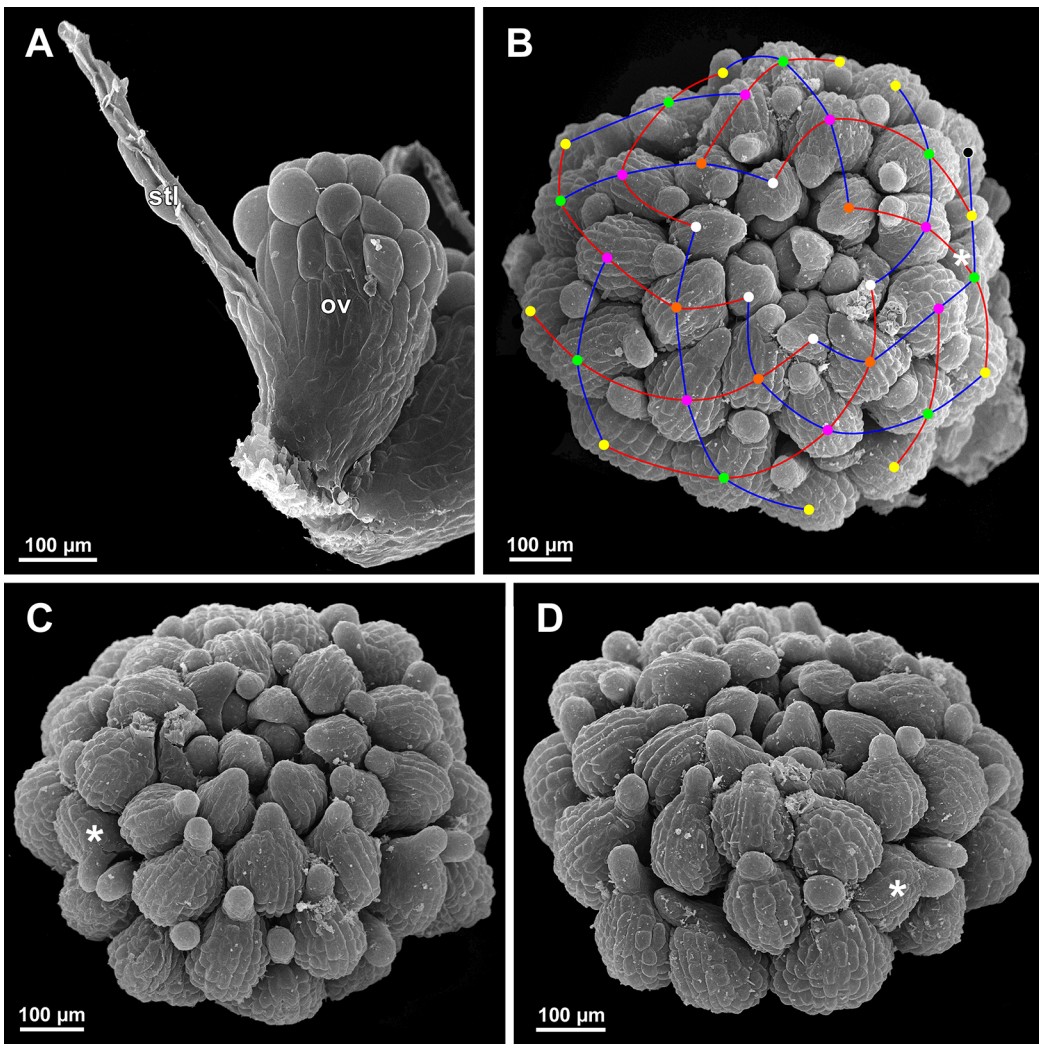

**Figure 29** *Sciaphila nana*, **development of gynoecium (SEM) (A, *Nuraliev 2445*, B–D: *Nuraliev et al. 972*).** (A) Carpel at style elongation, side view. (B) Gynoecium, top view, with superimposed parastichies of opposite directions indicated by red and blue lines; colour circles indicate carpel whorls; note the significant divergence of the parastichies with respect to the carpel number. (C and D) The same gynoecium as in B viewed from different angles, showing details of carpel arrangement. Labels: ov, ovary; stl, style. Asterisk indicates a carpel oriented at a right angle from its normal arrangement.

## DISCUSSION

### Groundplan of male flowers: number of whorls and their merism

Developmental data obtained for male flowers of *Sciaphila* suggest that the perianth is two-whorled, because the two alternating sets of tepals strongly differ in timing and manner of their initiation. Assuming that in nearly all monocots the floral whorls initiate in an acropetal sequence (*Rudall, 2010*; *Remizowa, 2019*; but see *Stützel, 1984*; *Vrijdaghs et al., 2009*), we interpret the early-arising tepals as an outer whorl, and the tepals with late-arising lobes as an inner whorl (Fig. 31). Similarly, in *S. densiflora*, the single species studied here with an androecium of six stamens, the androecium is also

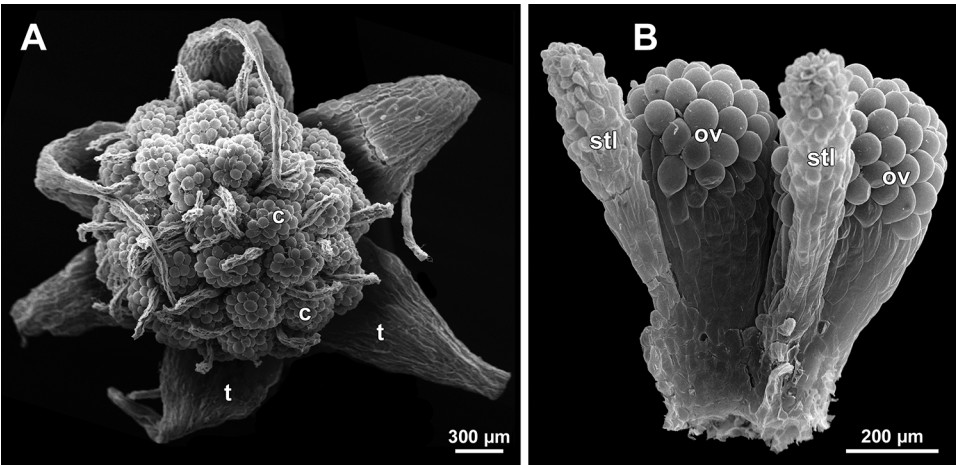

**Figure 30** *Sciaphila stellata*, **female flowers (SEM) (A,** *Nuraliev et al. 2499bis*, **B:** *Nuraliev & Kuznetsova 1380b*). (A) Anthetic flower, top view. (B) Two carpels, side view. Labels: c, carpel; ov, ovary; stl, style; t, tepal.

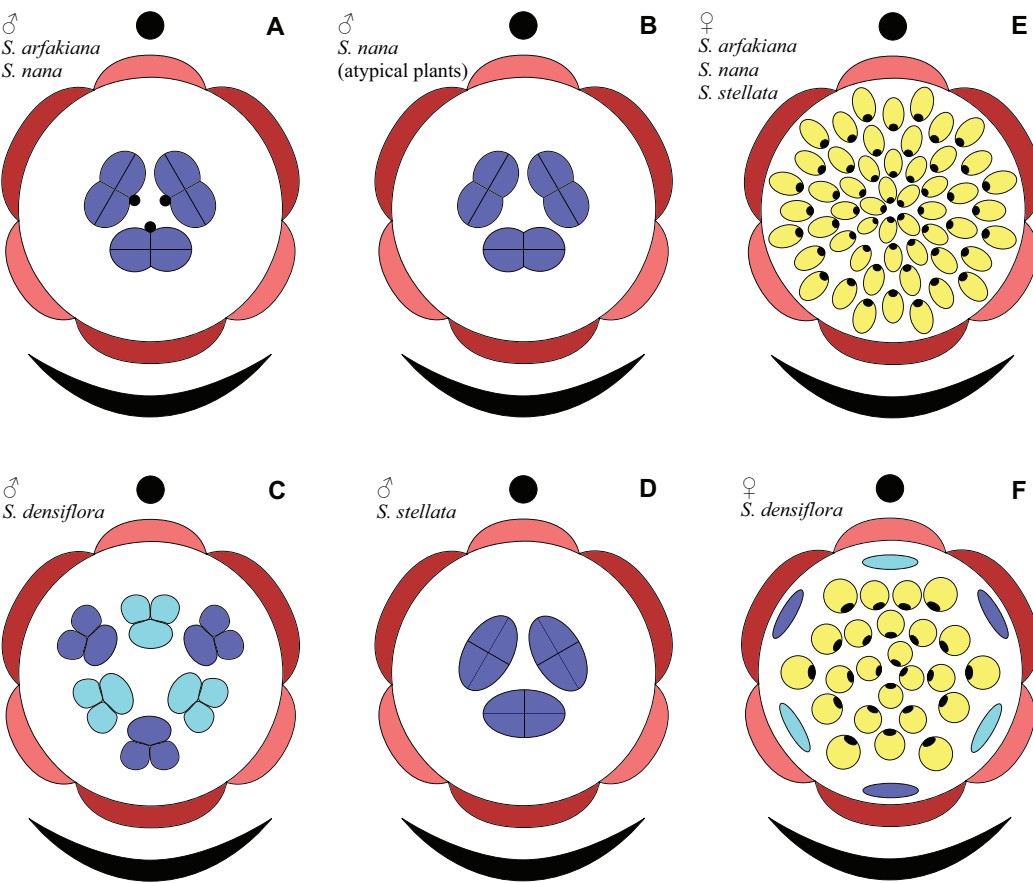

**Figure 31 Floral diagrams of studied species of** *Sciaphila* **(A–D) Male flowers. (E and F) Female flowers.** Dark red, outer tepals; light red, inner tepals; dark blue, outer stamens or staminodes; light blue, inner stamens or staminodes; yellow, carpels.

two-whorled, with the outer stamens in the radii of the outer tepals, and the inner stamens in the radii of the inner tepals. The two-whorled interpretation of the androecium in *S. densiflora* is supported by delayed initiation of the inner whorl of stamens. Therefore, in our interpretation, the male flowers of *S. densiflora* show a perfect alternation of all floral whorls, which is a basic pattern for angiosperms (*Endress, 2011*). Additionally, the abnormal flower of *S. nana* with four tepals and four stamens can also be assumed as having a two-whorled androecium, in contrast to the common single-whorled condition in this species.

Based on the idea of whorl alternation, we assume the three stamens in male flowers of *S. arfakiana*, *S. nana* and *S. stellata* to be homologous to the outer whorl of the androecium in *S. densiflora*. Thus, we describe the male flowers of *Sciaphila* with three stamens as tricyclic and those with six stamens as tetracyclic, differing in the absence or presence of the inner stamen whorl, with each floral whorl being typically trimerous (Fig. 31). This is consistent with the speculations by *Nuraliev, Cheek & Beer (2016)* who supposed two-whorled perianth and floral trimery in male flowers of *Seychellaria*, a genus of Triuridaceae closely related to *Sciaphila*.

The floral organs in the male flowers of *Seychellaria* which were treated as "filamentous structures" by *Rudall (2008)* and are commonly termed staminodes (see *Nuraliev, Cheek & Beer, 2016*) alternate with the stamens, and therefore their position in a flower fully corresponds to that of the inner stamens in six-stamened species of *Sciaphila* (such as *S. densiflora*). In our opinion, the criterion of position corroborates the androecial nature of these structures in *Seychellaria*. The male flower of *Seychellaria* is thus tetracyclic, with a remarkable stamen dimorphism among the androecial whorls.

Our results are in good agreement with illustrations of male flower development of *S. arfakiana* presented by *Rübsamen-Weustenfeld (1991*: Taf. 34–35*)*. *Rübsamen-Weustenfeld (1991)* interpreted the tepals of each whorl as developing sequentially, which is, however, not evident from her SEM images, which show all three primordia of a whorl being of nearly equal size at each stage. This interpretation also contradicts our findings that the tepals initiate simultaneously and develop with a delay at the abaxial side of a flower (see below). Although early perianth development was thoroughly studied and illustrated only for *S. arfakiana* (a single appropriate illustration provided for *Soridium spruceanum* Miers is quite inconclusive), *Rübsamen-Weustenfeld (1991)* extrapolated her data to assume the perianth of some other Triuridaceae as two-whorled, with the stamens occupying the radii of the outer whorl (see also *Maas-van de Kamer, 1995*; *Maas-van de Kamer & Weustenfeld, 1998*). These representatives include the neotropical *S. rubra* Maas characterized by a perianth of four tepals and androecium of four stamens. Our observations on two more species, *S. nana* and *S. densiflora*, support the idea that the two-whorled nature of the perianth is stable in *Sciaphila*.

Our observations do not support the hypothesis that tepals of Triuridaceae are arranged in pseudowhorls rather than true whorls (*Rübsamen-Weustenfeld, 1991*). The term "pseudowhorl" (*Charlton, 1973*; *Sattler & Singh, 1978*; *Posluszny & Charlton, 1993*) was introduced to describe patterns of organ arrangement in which phyllomes follow a spiral, but are closely spaced in groups superficially resembling whorls

(i.e., each phyllome is arranged in its own node). In our view, the perianth of *Sciaphila* follows the typical monocot groundplan with two trimerous whorls of tepals. We disagree with the view of *Rübsamen-Weustenfeld (1991)* that the early-arising tepals are initiated sequentially as 1, 2, 3 on the floral axis. In addition, sequential initiation of the phyllomes does not preclude their arrangement in a whorl, and this condition is well-documented in some monocots with typical flower groundplan (e.g. *Greller & Matzke, 1970*).

In both male and female flowers of *Sciaphila*, the tepals are known to be either equal or unequal, and in the latter case, the larger and smaller tepals alternate with each other (*Van de Meerendonk, 1984*). Since the perianth of *Sciaphila* (at least in the male flowers) is here shown to be two-whorled, a question arises for the species with dimorphic tepals regarding the correspondence between the tepal size (larger and smaller) and their attribution to the perianth whorls (outer and inner) in anthetic flowers. Perianth orientation with respect to the flower-subtending bract cannot be used for determination of the outer and inner tepals at anthesis, because the orientation becomes obscured due to pedicel elongation and torsion. The other way to distinguish tepal whorls in anthetic male flowers is to assess the positions of the tepals with respect to the stamens. In male flowers of *Sciaphila* which have unequal tepals and the stamens half as numerous as the tepals (i.e., two or three stamens), the stamens are known to occupy the radii of the larger tepals (*Van de Meerendonk, 1984*). From this uniform pattern it follows that the larger tepals are the outer ones and the smaller tepals are the inner ones in the flowers of this groundplan. However, in flowers with an equal number of tepals and stamens, like those of *S. densiflora*, the position of the stamens cannot be used for determination of perianth whorls, because each tepal is arranged in a stamen radius, and the stamen whorls are also indistinguishable. For such flowers, one can only suppose the same regularity as for the flowers with single-whorled androecium, but a possibility cannot be excluded that the inner tepals become larger than the outer tepals in the course of late developmental stages. Thus, for *S. densiflora* this question is still to be resolved by observation of subsequent stages of tepal growth till their maturity or possibly by comparative vascular anatomy.

## Patterns of perianth initiation in male flowers

We found that in the male flowers of *Sciaphila* the outer tepals are in a median abaxial and transversal-adaxial positions, and the two transversal-adaxial tepals grow considerably faster than the abaxial tepal. This type of floral orientation and early development is known to be common for lateral flowers in monocots which lack floral prophylls (bracteoles) (*Endress, 1995*; *Remizowa et al., 2013*). Accordingly, all species studied here were proved to be ebracteolate. The unidirectional nature of floral development is also evident from young inner tepals. At early stages, the perianth and the entire flower is thus prominently monosymmetric with a median plane of symmetry, and a shift to floral polysymmetry takes place when the tepals of each whorl become uniform in size. Floral polysymmetry with an early monosymmetric phase (also termed transient monosymmetry) has been reported to frequently occur in plants with spikes or racemes and to be often expressed by

delayed early development of the abaxial half of the flower (*Endress, 1999*, *2012*). It was supposed that the delay is caused by the influence of the flower-subtending bract, and several hypothetical mechanisms of this phenomenon have been suggested (*Endress, 1999*; *Remizowa et al., 2013*). In *Sciaphila*, the flower-subtending bract is much larger than the flower and covers it at the time of tepal initiation, which makes it possible that the developmental delay is caused by the physical pressure of the bract (*Ronse De Craene, 2018*). On the other hand, the influence of the bract through an inhibitory positional signal cannot be excluded. Notably, the developmental delay discussed here is not pronounced in the SEM images of *S. arfakiana* published by *Rübsamen-Weustenfeld (1991*: Taf. 34a–c*)*, but her work illustrates only three relevant flowers.

*Ambrose et al. (2006)* reported that "the first tepal primordium develops opposite the bract" in *Lacandonia schismatica* E. Martínez & Ramos. The illustration provided by *Ambrose et al. (2006*: fig. 9*)* shows an abaxial delay of perianth development similar to that in *Sciaphila*. In contrast to *Sciaphila*, the published image of flower of *L. schismatica* at the stage before gynoecium initiation (*Ambrose et al., 2006*: fig. 9) provides no evidence of the arrangement of the tepals in two whorls. Late developmental stages and anthetic flowers of both species of *Lacandonia* E. Martínez & Ramos show six equal tepals basally united in a tube (*Martínez & Ramos, 1989*; *Vergara-Silva et al., 2003*; *Ambrose et al., 2006*; *Rudall, Alves & Sajo, 2016*). Thus, the perianth of *Lacandonia* can be interpreted as either single-whorled or two-whorled. Notably, the perianth of *Triuris* Miers, the closest phylogenetic relative of *Lacandonia* (*Mennes et al., 2013*), consists of only three tepals (*Maas & Rübsamen, 1986*; *Vergara-Silva et al., 2003*) and is therefore clearly single-whorled.

*Sciaphila* reveals a highly unusual sequence of development of perianth parts: the outer whorl of tepals initiates as separate primordia, followed by formation of the perianth tube and finally by the initiation of free lobes of the inner tepals on the tube surface. Thus, the outer tepals demonstrate late congenital fusion, whereas the inner tepals exhibit early congenital fusion with adjacent outer tepals. The terms early and late congenital fusion (*Sokoloff et al., 2018*) are derived from the concept of early and late sympetaly (*Erbar, 1991*; *Leins & Erbar, 1997*, *2010*) and can be applied to describe a wide array of developmental processes in plants. The perianth tube itself appears to be of an intermediate (or perhaps dual) nature with respect to this feature in *Sciaphila*, as it is characterized by a combination of early and late syntepaly. A perianth tube of this type is apparently rare in angiosperms. A similar example is that of the calyx tube of *Coronilla* L. (Fabaceae), implying early synsepaly between two of the sepals and late synsepaly between the other three ones (*Sokoloff et al., 2007*, *2018*).

## Nature of the stamen appendages

The stamen appendages reported here for some species and specimens of *Sciaphila* are to be compared with so-called filamentous structures described in detail by *Rudall (2008)*, that is, the floral organs of Triuridaceae with an elongate shape, which are supposedly not homologous to any floral element or its part. Although *Rudall (2008)* did not mention the stamen appendages of *Sciaphila* among the filamentous structures, their homology is

not really obvious. As we described above, the appendage is attached close to the base of the stamen filament, and thus can be interpreted as either a part of stamen or an organ independently inserted on the receptacle. In our opinion, it is important here that we did not observe any variation in the number and arrangement of the appendages with respect to the stamens: there was constantly a single appendage associated with each stamen, and it occupied a strictly adaxial position. This contrasts with the relations of the stamens and tepals, which were found to infrequently vary in number and arrangement rather independently (see also *Rübsamen-Weustenfeld, 1991*; *Maas-van de Kamer, 1995*). For this reason, we treat the appendages as parts of the stamens.

Within this interpretation, the stamen appendages of *Sciaphila* and a closely related *Seychellaria* have been homologized with various parts of the stamen, including a distal portion of the filament (*Van de Meerendonk, 1984*) and an extension/appendage of anther connective (*Hemsley, 1907*; *Maas & Rübsamen, 1986*; *Rübsamen-Weustenfeld, 1991*; *Maas-van de Kamer, 1995*; *Maas-van de Kamer & Weustenfeld, 1998*; *Ambrose et al., 2006*; *Rudall & Bateman, 2006*; *Rudall, 2008*; *Merckx et al., 2013*). In fact, the appendage represents a basal adaxial outgrowth of the stamen filament, whereas the filament itself continues beyond the appendage attachment and is terminated by the anther.

It is noteworthy that the three fossil species described by *Gandolfo, Nixon & Crepet (2002)* in the genera *Mabelia* Gandolfo, Nixon & Crepet and *Nuhliantha* Gandolfo, Nixon & Crepet are characterized by true connective extensions (i.e., supraconnectives). The authors assumed connective extensions to be a common feature of these fossils and the appendage-bearing species of *Sciaphila* and *Seychellaria*, and used this similarity to support the attribution of the fossil taxa to Triuridaceae. In fact, the stamens of *Sciaphila* and *Seychellaria* show a different morphology and lack supraconnectives, as shown above. The supraconnectives are known to be completely absent from the extant Triuridaceae (*Maas-van de Kamer & Weustenfeld, 1998*), which is an argument against placement of *Mabelia* and *Nuhliantha* in this family. The affinities of the fossil flowers reported by *Gandolfo, Nixon & Crepet (2002)* with Triuridaceae have already been doubted on the basis of pollen characters (*Furness, Rudall & Eastman, 2002*; *Merckx, Bidartondo & Hynson, 2009*).

## Structure and development of carpels

In carpels of *Sciaphila* (*Rübsamen-Weustenfeld, 1991*; *Rudall, 2008*; this study), *Seychellaria* (*Rudall, 2008*) and other Triuridaceae where carpel development has been documented (*Ambrose et al., 2006*; *Rudall, 2008*; *Rudall, Alves & Sajo, 2016*), the ovules become discernible well before carpel closure. Late carpel closure (and early ovule development) was considered by *Endress (2015)* as a rare and apparently derived condition in angiosperms, found for example in derived families of Alismatales and in certain members of Poales, and also scattered among the eudicots, but unknown in any basal angiosperms. Notably, *Endress (2015)* has not listed any Pandanales in a review of cases of this condition. One can argue that the roundish structure in young carpels of Triuridaceae that we interpret as an ovule could be a short and thick adaxial wall of ascidiate zone of the carpel, whereas the ovule is formed after carpel closure

(as demonstrated for *Hopkinsia* Fitzg., Restionaceae, by *Fomichev et al. (2019)*. Special anatomical investigations of closed carpels at various developmental stages are needed to ultimately resolve this question. Currently we believe that the latter option is less probable, and this view is supported by illustrations of longitudinal sections of carpels of *Sciaphila* in *Rübsamen-Weustenfeld (1991*: Taf. 12d, e, 13f*)*.

*Endress (2015)* highlighted the difficulty in interpretation of free carpels characterized by a single median ovule and precocious ovule development. This assumption is fully applicable to *Sciaphila*. Indeed, the determination of carpel zones in *Sciaphila* is somewhat controversial, which is a result of peculiar carpel structure. At first glance, the ovary seems to be entirely formed by the folded carpel blade, with the ventral slit extending directly from the ovary base. This pattern of ovary development fits the idea of a plicate carpel with basal (but not dorsal or ventral) ovule attachment, lacking an ascidiate zone. However, at early stages of carpel development we observed a short ventral part of the carpel, and the ventral (but not basal) part is ovule-bearing. Initiation of the ovule closer to the ventral side cannot be interpreted in any other way than as an evidence in favor of ascidiate zone of the carpel (although very short). To summarize, we assume the carpel of *Sciaphila* as consisting of both ascidiate and plicate zones, the former being extremely short and undetectable in late development and in definitive carpels. The level of ovule attachment corresponds to the cross-zone within our interpretation. The ascidiate zone is hardly evident below the cross-zone. The plicate zone forms the rest of the carpel, that is, the major part of the ovary wall and the style.

The ascidiate zone is visible during an apparently short period of carpel development, and the features of carpel shape that indicate the presence of the ascidiate zone (the central depression and the flanges connecting the two carpel parts) are small and comparable to the carpel cells in their size. This leaves a possibility for argumentations contra our views. It is remarkable that the presence of an ascidiate zone is accepted in the earlier works which discussed the question of carpel structure in Triuridaceae (*Rübsamen-Weustenfeld, 1991*; *Igersheim, Buzgo & Endress, 2001*). Very similar gynoecium and carpel development (though without a gynobasic position of the style) have been reported for *Sagittaria* L. (Alismataceae) (*Huang, Wang & Wang, 2014*). Notably, the carpels of *Sagittaria* were shown to be capable of sharing the pollen by means of the growth of the pollen tubes through the receptacle, the latter thus functioning as an extragynoecial compitum (*Wang, Tao & Lu, 2002*). The arrangement of the ovules of Triuridaceae close to the receptacle tissues can possibly be related with the same mechanism. Pollen tube growth from anthers to the ovules through the receptacle has been reported to perform the pollination in the bisexual flowers of *Lacandonia* (*Márquez-Guzmán et al., 1993*; *Rudall, Alves & Sajo, 2016*).

Ovule position in *Sciaphila* can be compared to that in *Triuris*. *Triuris brevistylis* Donn. Sm. is known to have a highly similar carpel structure at early developmental stages, differing in the apparent attachment of the ovule to the receptacle (*Ambrose et al., 2006*: Fig. 54, 55). This phenomenon requires additional explanation, because angiosperm ovules are believed to be parts of carpels, and therefore are borne exclusively within their tissues (*Endress, 2019*). This inconsistency can be resolved by employment of the

notion (supported by histological data) that young carpels are "embedded" within the receptacle (*Van Heel, 1983*, *1984*; *Endress, 2019*). It thus can be accepted that the ovule of *T. brevistylis* is surrounded by the tissues of the ovary wall instead of being directly attached to the receptacle. These "embedded" carpel tissues surrounding the ovule possess an annular shape (in apical view), and can be treated as the basal part of the ascidiate zone.

The style in *Sciaphila* is entirely cylindrical, without a ventral furrow (see also *Igersheim, Buzgo & Endress, 2001*), and thus it seems to be unifacial, with the entire surface of the style formed by abaxial side of the carpel. However, in styles of *Sciaphila*, we observed no transitional area between bifacial and unifacial parts, which is usually present at bases of unifacial phyllomes (or their parts) (*Franck, 1976*). For this reason, there is a possibility that a narrow adaxial side is present throughout most of the style length. In the latter case, the style is to be described as a subunifacial (*Ozerova & Timonin, 2009*) structure. Apparently, *Endress (2015)* used the term "unifacial" to describe only the appearance of the style, but not its morphological nature, and implied a meaning of the cylindrical shape, opposed to the plicate condition. Both unifacial and hypothetical subunifacial types of the style differ from the plicate style in the absence of a ventral slit, and therefore of a postgenital closure.

### Gynoecium phyllotaxis

Our data on *S. arfakiana* show a high variation in patterns of arrangement of the outermost carpels with respect to the tepals, with all possible options detected: a carpel in a tepal radius, a carpel in an intertepal radius, two carpels in a tepal sector. Nevertheless, we found a tendency to the paired arrangement of carpels in tepal sectors. In both available flowers clearly showing six tepals and twelve carpels, such a pattern was manifested all round. In cases with this feature investigated only in a floral sector, the pairwise arrangement was found together with the other types of arrangement.

In *S. densiflora*, the staminodes occupy the tepal radii and thus are most likely arranged in two whorls by analogy of the fertile stamens of the male flowers of the same species, at least in the flower with six staminodes (Fig. 31). The outermost carpels of *S. densiflora* alternate with the staminodes (though this is based on study of a single flower). One can suppose that the presence of the staminodes plays an important role in the transference of the positional information from perianth to gynoecium in *S. densiflora*, and this positional signal is significantly weaker in flowers of *Sciaphila* lacking the staminodes.

The three flowers of *S. nana* used for the investigation of the phyllotaxis of the entire gynoecium show a remarkable diversity of carpel arrangement. One of the flowers possessed regular parastichies and whorled structure of the gynoecium, with all whorls being 9-merous except for the two distal ones. The other flower had moderately smooth parastichies and the numbers of parastichies of opposite directions were inferred as 10 and 11. This can be seen as indicative of a special kind of spiral phyllotaxis with divergence angle of c. 34.3° (rather than c. 137.5° of the Fibonacci spiral). Alternatively, this can be viewed as a whorled system with non-integer merism, namely, a merism of 10.5 in several basal whorls. The choice between these two interpretations is rather

conventional, as discussed by *El, Remizowa & Sokoloff (2020)* using examples from androecia of *Nuphar* (Nymphaeaceae). The third flower of *S. nana* examined here in detail is characterized by rather a chaotic carpel arrangement that cannot be assigned to accurate sets of parastichies. Notably, none of these flowers possesses a gynoecium with right and left sets of parastichies differing in more than one parastichy, that would unambiguously point to the spiral phyllotaxis. However, the studied examples demonstrate that it is still difficult to provide an unequivocal characteristic of carpel phyllotaxis for *S. nana* and the genus *Sciaphila* in general. Rather, we suggest to interpret this feature in the studied plants as a result of a balance between a particular kind of order and an irregular arrangement.

In a given flower, the balance seems to be shifted to one or the other side, probably depending of the conditions of the development of the flower (such as exact size of the floral apex and the tepals at the time of carpel initiation or prepatterning). The degree of irregularity of floral phyllotaxis is known to be positively correlated with the number of involved organs in angiosperms (*Endress & Armstrong, 2011*; *Rutishauser, 2016*; *El, Remizowa & Sokoloff, 2020* and references therein), and thus the irregular patterns in carpel arrangement in *Sciaphila* are quite expected. The opposite tendency, that is, an ordered phyllotaxis, can possibly be inherited by a gynoecium from the preceding floral whorl through the positional signal produced by the perianth and/or androecium. This idea is consistent with the merism of the outermost whorl of carpels which is often divisible by three (Fig. 31), as assessed from diagramming of parastichies (*S. nana*: 9, 10 and possibly 10.5 carpels) and from direct counting of primordia at early stages of gynoecium development (*S. arfakiana*: usually 12 carpels, also 9 and 11 carpels; *S. nana*: 9 carpels). In contrast to the staminodes which are supposed, as discussed above, to govern rather strongly the position of the basalmost carpels in *S. densiflora*, the tepals seem to possess a relaxed control over the number and position of carpels. The precise patterns of transference of the positional information from the tepals to the carpels are either highly variable or species-specific: we observed a tendency to the presence of twelve outermost carpels in *S. arfakiana* and nine in *S. nana*. Both these figures are likely to be somehow related to the trimerous nature of the two perianth whorls, but imply different carpel arrangement with respect to the tepals. We interpret the carpel whorls with merism that is not divisible by three in *Sciaphila* as formed by a significant contribution of irregularity.

Our interpretation of gynoecium phyllotaxis in *Sciaphila* generally agrees with those of *Rudall (2008)* for *Sciaphila* and *Seychellaria*. *Rudall (2008)* described the carpel arrangement in *Sciaphila major* Becc. and *S. tenella* Blume as "somewhat chaotic". She characterized the carpel arrangement in *Seychellaria madagascariensis* C.H. Wright and *S. thomassetii* Hemsl. as "spiral or chaotic", and noted that the numbers of right and left parastichies in a gynoecium is apparently the same, which indicates, according to Rudall, "a whorled arrangement, although this becomes chaotic in places". Thus, *Rudall (2008)* did not propose a conclusion for this question, which is readily understandable in the light of the broad variation uncovered in our investigation. It is remarkable that a gynoecium of *Seychellaria thomassetii* is coloured as having twelve parastichies (*Rudall, 2008*: Fig. 10B).

This gynoecium could be 12-merous, a condition we also found in *S. arfakiana*, but this hypothesis should be tested by drawing a set of parastichies of the opposite direction.

Patterns of carpel arrangement in *Sciaphila* and *Seychellaria* outlined above allow questioning the interpretation of gynoecium phyllotaxis as spiral in *Kupea martinetugei* Cheek & S.A. Williams, a member of the tribe Kupeaeae (*Rudall et al., 2007*). It will be essential to clarify the occurrence of whorled or spiral carpel arrangement in *Kupea* Cheek & S.A. Williams by mapping positions of individual carpels as performed here for *S. arfakiana* and *S. nana*. It is possible that the gynoecium of *Kupea* is whorled or irregular as in *Sciaphila*.

## Structure of inflorescence in *Sciaphila* and *Seychellaria*

*Rübsamen-Weustenfeld (1991*; see also *Maas-van de Kamer & Weustenfeld, 1998)* has summarized that the entire family Triurudaceae is characterized by racemose inflorescences, with the exception of most species of *Seychellaria*. In *Seychellaria thomassetii*, the inflorescence is also a raceme, whereas other species of this genus show diverse deviations in inflorescence structure, as outlined by *Rübsamen-Weustenfeld (1991)* and described in detail by *Nuraliev, Cheek & Beer (2016)*. *Seychellaria madagascariensis* possesses a thyrse, with a cyme of up to four flowers in an axil of each phyllome of the main axis. The flowers of *S. madagascariensis* are known to be bracteolate, with usually two bracteoles per cyme, and thus the presence of the bracteoles in the flowers of higher orders of the cyme is still questionable. *Seychellaria africana* Vollesen has racemes with bracteolate flowers; this condition was regarded as intermediate between a raceme and a thyrse by *Rübsamen-Weustenfeld (1991)*, see also *Remizowa et al. (2013)*. A similar inflorescence structure was found in the recently described species *Seychellaria barbata* Nuraliev & Cheek, with the difference in the presence of underdeveloped flowers in the axils of bracteoles, which makes the inflorescence more close to a thyrse than to a raceme (*Nuraliev, Cheek & Beer, 2016*). Additionally, the bracteoles of some species of *Seychellaria* are laterally united with flower-subtending bracts (*Nuraliev, Cheek & Beer, 2016*), which complicates clarification of the morphological nature of bracteole and the inflorescence structure in this genus.

Our investigations in *Sciaphila* confirm the idea of complete absence of bracteoles and the truly racemose nature of inflorescence in the genus. Given that *Sciaphila* is likely to be paraphyletic with respect to *Seychellaria* (*Mennes et al., 2013*), we suggest that the cymose partial inflorescences of *Seychellaria* have appeared in the course of evolution from the single axillary flowers of *Sciaphila*, and the appearance of the bracteoles acted as a key innovation that allowed inflorescence branching.

*Maas-van de Kamer (1995)* indicated, apparently based on the study by *Rübsamen-Weustenfeld (1991)*, that in the entire family Triuridaceae the median tepal of the outer whorl is always arranged abaxially (facing the flower-subtending bract). Indeed, it is likely to be so for all the representatives of the family with racemose inflorescences and lacking bracteoles, and it was observed in the species of *Sciaphila* studied here. However, we suppose that all the species of *Seychellaria* with bracteolate flowers possess variable floral orientation, as it has already been reported for *Seychellaria barbata*

(*Nuraliev, Cheek & Beer, 2016*) and found to be common in other monocots which possess a single floral prophyll (*Remizowa et al., 2013*).

## Euanthial interpretation of reproductive units of Triuridaceae in the light of new data

The reproductive units of Triuridaceae, traditionally termed flowers, were stressed to show certain features of inflorescences by *Rudall (2003*, *2008)* and *Rudall & Bateman (2006)*. Their suggestion of a pseudanthial interpretation stimulated comprehensive studies of these structures. As a result, the reproductive units of *Lacandonia* and *Triuris* were concluded to be homologous to flowers of the other angiosperms (*Ambrose et al., 2006*).

Our results allow consideration of the reproductive units of *Sciaphila*, the largest genus of Triuridaceae. In *Sciaphila*, structure, orientation and developmental patterns of the reproductive units appear to closely meet the groundplan of a typical lateral monocot flower lacking a bracteole (*Endress, 1995*; *Remizowa, Sokoloff & Rudall, 2010*; *Remizowa et al., 2013*). The male flowers of *Sciaphila* are commonly trimerous, with a two-whorled perianth, one- or two-whorled androecium and a precise alternation of all the floral whorls. Perianth development is abaxially delayed in the same way as in other monocots with lateral flowers lacking a bracteole. The female flowers are also six-tepaled. The morphologically bisexual flowers (including functionally bisexual and staminode-bearing) show sequentially arranged perianth, androecium and gynoecium, each corresponding to the same floral zone of a unisexual flower in number and arrangement of the elements. Thus, the available data on *Sciaphila* supports the idea of the euanthial nature of the reproductive units in the entire family Triuridaceae.

## Taxonomic implications

*Problematic assessment of anther structure in the genus Sciaphila*

Anther morphology is widely used for taxonomic segregation and identification of species of *Sciaphila*. The anthers are believed to be uniformly dithecal in *Sciaphila* (*Maas & Rübsamen, 1986*; *Merckx et al., 2013*). *Endress & Stumpf (1990)* characterized the anthers of Triuridaceae (exemplified by *Lacandonia*, *Sciaphila* and *Soridium* Miers) as dithecal but at the same time synthecal and di- or trisporangiate, which is partly in contradiction with our findings and earlier investigations. Particularly, the anthers of *Sciaphila* are here proved not to be synthecal, but there is still a possibility that synthecal anthers are present in *Soridium* (*Maas & Rübsamen, 1986*: Figs. 15B and 15D). Clearly, the number of thecae in anthers of most Triuridaceae remains an open question, because they lack a pronounced connective, which makes them similar to monothecal anthers irrespective of the number of microsporangia.

Two variable characters are mainly used in taxonomy of *Sciaphila*, viz. the number of locules (= microsporangia, cells) and the number of lobes. Apparently, the number of locules and number of lobes are not necessarily equal in a given anther: for example, the anthers are stated to vary in the genus within 1–4-locular and 2–4-lobed by *Van de Meerendonk (1984)*, who, however, has not indicated number of anther locules for most of the species in his account. In contrast to *Van de Meerendonk (1984)*, other authors

(*Maas-van de Kamer, 1995*; *Maas-van de Kamer & Weustenfeld, 1998*; *Merckx et al., 2013*) described the anthers of *Sciaphila* as 3–4-locular and mentioned 2-locular anthers only for some other genera of Triuridaceae.

Whereas the 3-locular condition seems to be easily recognizable (and apparently coincides with 3-lobed shape), evaluation of the 2- vs. 4-locular (and -lobed) condition is not always straightforward. In our investigation, it is demonstrated by the example of *S. stellata*. All the studied male flowers of this species possess 2-lobed (more precisely, 2-valved) anthers that strongly resemble 2-locular anthers, including the closed and dehisced ones. Most likely, it is the presence of two valves around a seemingly single cavity that led *Averyanov (2007)* to indicate the anther structure of *S. stellata* as unilocular. However, our investigation of the internal structure of an anther uncovered the presence of four locules. Thus, in 2-lobed anthers of *Sciaphila* the number of locules cannot be established with confidence by external observations alone. It means that the complete absence of 2-locular anthers in *Sciaphila* is highly probable, and the reports of such anthers in the genus (e.g. *Xu, Li & Chen, 2011*; *Suetsugu et al., 2017*; *Suetsugu, Kinoshita & Hsu, 2019*; *Suetsugu & Kinoshita, 2020*) are possibly misinterpretations and require careful verification. The case of *S. secundiflora* Thwaites ex Benth., a species morphologically very similar to *S. stellata* (see below), is particularly illustrative: the anthers of this species were described as 4-locular and 4-lobed by *Van de Meerendonk (1984)* but stated to be 2-locular by *Suetsugu, Kinoshita & Hsu (2019)*.

Although 2-locular anthers are probably absent from *Sciaphila*, the remarkable diversity in anther shape could still have taxonomic significance in this genus. In order to make this character clear and unequivocally applicable, it is essential to precisely define the difference between the 2-lobed and 4-lobed conditions, that is, what is termed by the anther lobe for both closed and dehisced anthers.

*Taxonomic status of Sciaphila stellata*

*Sciaphila stellata* was described on the basis of two specimens collected in northern Vietnam (*Averyanov, 2007*). Since description, no more findings of this species have been reported, except for a specimen from Guangxi (China) (*Jiang et al., 2011*), which is very close geographically to the type location of the species, but the illustrations of this plant (along with the general taxonomic uncertainties regarding *S. stellata*) do not allow us to verify the identification.

*Sciaphila stellata* belongs to a taxonomically difficult complex of species, in which the earliest described species is *S. secundiflora*. The protologue of *S. stellata* notes the similarity of this species with *S. secundiflora*, but does not indicate explicitly the differences between them (*Averyanov, 2007*). The comparison is currently complicated by the lack of a commonly accepted species concept for this group. Within the treatment by *Van de Meerendonk (1984)*, *S. secundiflora* is an extremely variable and widespread species (with a large number of heterotypic synonyms), and the original material of *S. stellata* (the type and the paratype) along with the specimens studied here would certainly fit this circumscription. Some other authors, in contrast, tend to accept the taxa of *S. secundiflora* complex as distinct species. Particularly, *Suetsugu, Kinoshita & Hsu (2019)*

have reinstated two species from the synonyms of *S. secundiflora* using differences in size of some floral parts. The latter approach is difficult to follow, because some of the measurements are unknown for type specimens of some species from this complex (including *S. stellata*), and because the distinctness of the segregate species has never been tested by a quantitative analysis of a relevant amount of material.

Our identification of the specimens treated here under *S. stellata* was based on their remarkable similarity with the description and the images from the protologue of this species and on the relative geographical proximity of our specimens to its type location. The known populations of *S. secundiflora* complex geographically closest to our specimens are those of *Sciaphila stellata*, although they are found in another part of Vietnam. We argue that comprehensive morphological investigation is crucial for further clarification of taxonomic boundaries within this species complex, and the detailed descriptions verified by SEM images would allow accurate documentation of plant structure and avoid such uncertainties as the discrepancy in anther locules of *S. secundiflora* pointed above.

*Striking morphological diversity of Sciaphila nana*
Among the studied specimens of *S. nana*, the specimen *Nuraliev et al. 972* differs remarkably from the others in several aspects of the male flowers. First, it showed a nearly globose knob at the apex of the inner tepals, which is sharply delimited from the narrow portion of tepal (vs. cylindrical, smoothly delimited knob). Second, the stamens of this specimen completely lack the appendages (vs. prominent appendages, usually much longer than the anthers). Additionally, the specimen *Nuraliev et al. 972* possesses shorter male pedicel (± as long as flower vs. much longer than the flower) and somewhat larger male flowers at anthesis (ca. 3 mm vs. ca. 1.5–2 mm in diameter when perianth lobes spread; this is to be verified by more numerous measurements). The specimen in question was found within several kilometers of another specimen of *S. nana*, *Nuraliev 498*, that has "typical" morphology (as accepted here). However, the species of *Sciaphila* frequently occur in mixed populations, with sometimes just a few meters between the individuals of different species (as is evident from the other specimens studied here).

Both morphotypes outlined above generally fall within the current morphological concept of *S. nana* (*Van de Meerendonk, 1984*; *Chantanaorrapint & Thaithong, 2004*; *Averyanov, 2007*). *Sciaphila nana* is known to be characterized by significant morphological variation; moreover, it was suggested to unite this species with *S. arfakiana* (*Guo & Cheek, 2010*), which would make it even more variable, but this view has never been adopted and is not supported by our study. Particularly, stamen appendages are stated to be often present in *S. nana* in Flora Malesiana (*Van de Meerendonk, 1984*, "filaments often exceeding the anthers") and are apparently absent from the Thai specimen assigned to this species (*Chantanaorrapint & Thaithong, 2004*). *Ohashi et al. (2008)* described *S. nana* as having stamens with "a minute cylindrical appendage" and also its male flowers as having three reduced carpels, a feature never mentioned elsewhere for this species. Short but prominent appendages are reported for the Korean populations of *S. nana* (*Yim, Kim & Song, 2011*). The stamen appendages are absent from the

Myanmarese specimen of the species (*Jin, Zhu & Mint, 2018*). *Averyanov (2007)* stated "filaments hardly exceeding anthers" for *S. nana*; notably, his specimens listed under this species were collected in the same location as our "unusual" specimen *Nuraliev et al. 972* (Chu Yang Sin National Park), and their photographs (*Averyanov, 2007*: Fig. 2C and 2D) are highly similar to it.

The remarkable diversity of Asian specimens of Triuridaceae with respect to the absence or presence of the stamen appendages, that substantially contributes to the appearance of the entire flower, has been acknowledged long time ago. The plants with appendaged stamens were formerly assigned to the genus *Andruris* Schltr., as opposed to *Sciaphila* s.s. which was stated to lack the appendages (*Schlechter, 1912*; *Giesen, 1938*). As shown later, *Andruris* and *Sciaphila* s.s. are indistinguishable in other respects; consequently, *Andruris* is currently being synonymized with *Sciaphila* (*Gandolfo, Nixon & Crepet, 2002*; *Hsieh, Wu & Yang, 2003*; *Ohashi et al., 2008*; *Govaerts, Maas-van de Kamer & Maas, 2020*; but see *Maas-van de Kamer & Weustenfeld, 1998*; *Merckx et al., 2013*), and furthermore, some of its species are treated as heterotypic synonyms of certain species of *Sciaphila*, including a great number of them under *S. arfakiana* (*Van de Meerendonk, 1984*). Thus, within the current views, this feature frequently varies at intraspecific level in *Sciaphila*.

Our preliminary conclusion is that two distinct morphotypes are indeed present in Vietnam, but the differences are insufficient (at least, at the current state of the knowledge) to treat them as separate taxa. The alternative option can become more plausible when a special comprehensive investigation is carried out that should include consideration of the corresponding characters of the type of *S. nana*. The occurrence of evolutionary lineages within this species and their congruence with the morphotypes and geographical distribution is to be tested by molecular phylogenetic data.

*Sciaphila densiflora: nuances of identification and a new record for Vietnam*
The specimen *Nuraliev 1670* treated here as *S. densiflora* shows a high similarity with *S. arcuata* Aver. *Sciaphila arcuata* is a Vietnamese endemic which is known only from the type gathering collected 287 kilometers from the location of the specimen *Nuraliev 1670* (*Averyanov, 2007*). The protologue of this species states that the male flowers are unknown, and the identity of *S. arcuata* is quite obscure for the reason that male flowers contain most of the characters necessary for comparison with the other species (*Van de Meerendonk, 1984*). During investigation of the isotype of *S. arcuata* (LE: LE01041991) we have found that there are several flower buds in the terminal portion of the inflorescence, which are almost definitely male according to the inflorescence structure common in the genus. However, destructive sampling of this collection is problematic due to its unique nature.

Within the treatment in Flora Malesiana (*Van de Meerendonk, 1984*), which also covers most of the species of *Sciaphila* known in the mainland Southeast Asia, *S. arcuata* is most similar, in our opinion, to the widespread *S. densiflora*. The protologue (*Averyanov, 2007*) contains a brief comparison of *S. arcuata* with *S. micranthera* Giesen, but does not mention *S. densiflora*. *Sciaphila arcuata* shares with both *S. densiflora* and

*S. micranthera* a non-secund inflorescence (i.e., with flowers arranged all around and not turned to one side) and bearded tepals of female flowers. However, *S. arcuata* is closer to *S. densiflora* in the number of carpels in a gynoecium: there are 45 carpels illustrated in the drawing of *S. arcuata* (*Averyanov, 2007*: Fig. 1D), ca. 15–40 carpels in *S. densiflora* and ca. 15–25 carpels in *S. micranthera* (*Van de Meerendonk, 1984*). As is evident from the protologue and the isotype of *S. arcuata*, it differs from *S. densiflora* in subequal (vs. conspicuously unequal) and triangular-ovate (vs. long-triangular) free parts of female tepals and in the attachment of the style subapically on the carpel (vs. at the base or middle of the carpel). Therefore, we consider *S. arcuata* as a distinct, although poorly known, species until more evidence on its morphology and phylogenetic placement become available.

The specimen studied here (*Nuraliev 1670*) nearly perfectly fits the description and illustration of *S. densiflora* provided by *Van de Meerendonk (1984)* with respect to the inflorescence, male and female flowers. At the same time, it fails to fit *S. arcuata* by its pronouncedly unequal, long-triangular lobes of female tepals, the longer ones with a caudate apex and all of them lacking an apical knob, and by distinctly basal attachment of the style. Thus, we assign our specimen to *S. densiflora* and report a significant range extension for this species, which has earlier been known to inhabit Sri Lanka, Borneo, the Philippines, the Lesser Sunda Islands, the Maluku Islands, New Guinea and New Caledonia (*Van de Meerendonk, 1984*; *Tsukaya & Suetsugu, 2014*) and is presented here as a new record for Vietnam and for the entire mainland Asia.

*Staminodes in Sciaphila densiflora and the uncovered similarity of S. densiflora with S. tenella*

The flowers of *S. densiflora* (including *S. flexuosa* Giesen, *S. longipes* Schltr., *S. nutans* Giesen, *S. reflexa* Schltr., *S. trichopoda* Schltr.) are known to be entirely unisexual, that is, staminate and carpellate flowers without any morphological structures of the opposite sex (*Schlechter, 1912*; *Giesen, 1938*; *Van de Meerendonk, 1984*). Here we describe for the first time the female flowers of this species as morphologically bisexual (functionally unisexual), according to our observation that they possess the staminodes. Although these structures show very simple shape and lack any specific androecial features, we consider them to be homologous to stamens on the basis of their number and arrangement which correspond with the stamens of male flowers of *S. densiflora*. Furthermore, if the presence of staminodes in *S. densiflora* is considered, the groundplan of the female flowers of this species appears to be identical to that of bisexual flowers of another widespread Asian species, *S. tenella* (*Giesen, 1938*; *Van de Meerendonk, 1984*; *Ohashi et al., 2008*; *Guo & Cheek, 2010*; *Chantanaorrapint & Chantanaorrapint, 2012*). This can be viewed as additional support for a staminodial interpretation of the sterile organs of *S. densiflora* in question.

Apart from the bisexual flowers, *S. tenella* possesses male flowers which are of the same structure as those of *S. densiflora*, including the peculiar 3-locular anthers. In total, it appears that the presence of (functionally) female vs. bisexual flowers is the only character that clearly delimits *S. densiflora* from *S. tenella*. The other possible difference is the

filaments that are sometimes described as basally connate for *S. tenella* (*Van de Meerendonk, 1984*), but this feature is most likely indistinct and variable, especially if *S. maculata* Miers is treated as the synonym of the latter species, as suggested by *Ohashi et al. (2008)* and accepted by *Govaerts, Maas-van de Kamer & Maas (2020)*. Our finding of morphologically bisexual flowers in *S. densiflora* leads to the understanding of its closest similarity to *S. tenella*, and brings us to a question, if they indeed represent two distinct species or should be merged into one species. The latter option would be in agreement with the known morphological variability already documented for both species as well as with the geographical evidence. The question is to be resolved by investigation of morphology of greater amount of specimens and molecular phylogenetic studies.

A more general speculation can be suggested based on our results on *S. densiflora*, that is, that the morphologically bisexual flowers in *Sciaphila* are more widespread than is currently documented. In *S. densiflora*, the staminodes are very small (much smaller than the carpels), completely hidden between the perianth and the gynoecium and are undetectable without special examination. Therefore, similar staminodes may be overlooked in some other species of *Sciaphila* with female flowers (though we have confirmed their absence in three other species studied here). In support of this hypothesis, there is an example provided by *Rübsamen-Weustenfeld (1991)* who has observed bisexual flowers in the neotropical *S. albescens* Benth. which was previously known to be strictly monoecious.

## CONCLUSIONS

Investigations of Triuridaceae, especially those dealing with taxonomy and evolutionary morphology, have for a long time been complicated by the scarcity of information about floral groundplan and structure of the floral organs of their representatives. We have clarified the floral morphology in four species of *Sciaphila* and discussed data on other members of the family in the context of our findings.

Male flowers of *Sciaphila* are (usually) trimerous and tricyclic or tetracyclic depending on the number of stamens. Male flowers of the closely related genus *Seychellaria* are tetracyclic, with the inner androecium whorl being represented by staminodes. The outer tepals in the male flowers of *Sciaphila* are in a median abaxial and transversal-adaxial positions, as in other monocots with the same type of inflorescence. The perianth development demonstrates an abaxial delay during early stages, and thus the flower is polysymmetric with early monosymmetry. The phenomenon of unidirectional development has been reported for many angiosperms (including monocots) with relatively large flower-subtending bracts and symmetrical arrangement of floral prophylls (e.g. *Endress, 1995*; *Remizowa et al., 2013*). In *Sciaphila*, the perianth tube arises after the outer tepals but before the free lobes of the inner tepals, and therefore the outer tepals are characterized by late congenital fusion, whereas the inner tepals show early congenital fusion. The stamen appendages are adaxial outgrowths of the stamen filaments. Anthers are 3- osr 4-locular in *Sciaphila*, and reports of 2-locular anthers in this genus resulted from diversity of external appearance of anthers, but the external appearance is sometimes misleading.

The carpels of *Sciaphila* consist of a very short ascidiate zone and a massive plicate zone. The ovule is attached in the cross-zone. The carpel arrangement in gynoecium of *Sciaphila* combines the patterns of whorled and irregular phyllotaxis. The number of carpels in a whorl is estimated as 6 (or 7) in *S. densiflora* and 9, 10, 10.5, 11 or 12 in *S. arfakiana* and *S. nana*, with the prevalence of the numbers that are divisible by three. There are no unequivocal indications of spiral carpel arrangement in *Sciaphila*.

We report *S. densiflora*, which has never been listed for mainland Asia, as a new record for Vietnam. We describe the carpel-bearing (female) flowers of this species as morphologically bisexual according to our finding that they possess small staminodes.

Our study provides an important basis for investigation of floral evolution in Pandanales by means of character state reconstruction. It is a large step towards understanding how the morphological misfit family Triuridaceae has evolved. Particularly, the study opens a possibility for precise attribution of character states to the main floral traits of *Sciaphila* and some other genera of Triuridaceae.

We have uncovered a number of morphological similarities and differences between the species of *Sciaphila*, which indicate the necessity of verification of distinctness of several taxa among its Asian representatives. The presence of staminodes in female flowers of *S. densiflora* makes this species hardly distinct from *S. tenella*. *Sciaphila arcuata* belonging to the same morphological group is a poorly known but probably separate species. The Vietnamese samples of *S. nana* show remarkable heterogeneity, and are represented by two morphs differing in pedicel length, shape of tepal knobs and absence or presence of stamen appendages. Finally, a revision of the entire *S. secundiflora* species complex is needed to clarify the identity of the doubtful *S. stellata*. These taxonomic uncertainties are to be resolved by molecular phylogenetic investigations.

## ACKNOWLEDGEMENTS

We are grateful to Andrey N. Kuznetsov, Svetlana P. Kuznetsova and Eugene S. Popov for field assistance, to Leonid V. Averyanov for his help with investigation of type specimens of *Sciaphila* in LE and to Kenji Suetsugu for his ideas on species identity of the studied plants. We thank Paula Rudall and Martin Dančák for their comments and suggestions.

### Funding

The study is supported by the Russian Science Foundation (project 19-14-00055). The funders had no role in study design, data collection and analysis, decision to publish, or preparation of the manuscript.

### Grant Disclosures

The following grant information was disclosed by the authors:
Russian Science Foundation: 19-14-00055.

## Competing Interests

The authors declare that they have no competing interests.

## Author Contributions

- Maxim S. Nuraliev conceived and designed the experiments, performed the experiments, analyzed the data, prepared figures and/or tables, authored or reviewed drafts of the paper, and approved the final draft.
- Margarita V. Remizowa conceived and designed the experiments, performed the experiments, analyzed the data, prepared figures and/or tables, authored or reviewed drafts of the paper, and approved the final draft.
- Dmitry D. Sokoloff conceived and designed the experiments, performed the experiments, analyzed the data, prepared figures and/or tables, authored or reviewed drafts of the paper, and approved the final draft.

## Field Study Permissions

The following information was supplied relating to field study approvals (i.e., approving body and any reference numbers):

Permits were provided by the Vietnamese Government with participation of the Russian-Vietnamese Tropical Center: 547, 1951, 308.

## Data Availability

The raw data is presented in Figs. 1–30.

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
