# Peer review of "Flower structure and development in Vietnamese Sciaphila (Triuridaceae: Pandanales): refined knowledge of the morphological misfit family and implications for taxonomy"

_PeerJ, doi:10.7717/peerj.10205_

## Round 0.1 · original submission · Minor Revisions

Both reviewers agree that only minor revisions are needed to the manuscript. Please revise the manuscript based on the reviewers' comments, and provide a rebuttal letter explaining the changes made and a tracked changed manuscript.

·

Basic reporting

This is an excellent paper that merits publication with little revision. It provides descriptions of developmental morphology in two genera of the mycoheterotrophic family Triuridaceae. This unusual family is of broad general interest but is highly challenging to collect and examine, and hence is relatively poorly-known. I particularly like the attention to detail, such as the record of a “globose knob” (which appears glandular) at the tips of the tepals in one unusual specimen of S. nana.

The paper is well presented with high-quality images. There are currently 30 figures, each containing few images, all of them essential for the paper. The logistics of embedding this number of figures into a relatively short text might be problematic, so if possible I would suggest combining some of them into fewer but larger figures (but not essential).

In general, the english is good, but there are a few typos and other grammatical issues that I have marked separately on an uploaded pdf file.

Experimental design

No comment

Validity of the findings

No comment

Additional comments

Flowers versus inflorescences: The question regarding the evolutionary derivation and homologies of the reproductive units of Triuridaceae (Introduction, lines 50-53) is largely a matter of interpretation and has neither been proved nor refuted (lines 53-54). Rudall, Alves & Sajo, (2016) used the term “flower”, but pointed out that the homeosis theory (Vergara-Silva et al., 2003; Ambrose et al., 2006; Álvarez-Buylla et al., 2010; etc) was applied only to the inside-out flower of Lacandonia. The homeosis theory therefore does not address the question whether the Triuridaceae flower is evolutionarily derived from a flower or an inflorescence, because homeosis (heterotopy) in Lacandonia occurred well after the family Triuridaceae evolved (see Fig. 13 in Rudall, Alves & Sajo, 2016).

Filamentous structures: The term “morphological novelty” does not necessarily apply to organs sui generis (lines 57-58); Rudall (2008) applied this term to several unusual features in triurids, including the presence of carpel fascicles, centrifugal carpel inception, paired carpel arrangement, unusual filamentous structures in atypical positions, and the remarkable inside-out flowers of Lacandonia. The fact that these unusual structures are associated with different organs in different triurids is remarkable (as nicely highlighted in this paper).

Tribal classification: There is currently no mention of the existing tribal classification within Triuridaceae. I would suggest at least mentioning that Sciaphila belongs to the tribe Sciaphileae Miers (five genera: Merckx et al. 2013).

Generic limits: As the authors state, it is entirely reasonable to continue to recognise Sciaphila and Seychellaria as distinct from each other pending further data, as the analysis of Mennes et al. (2013) did not include all species and encompassed some long branches. Mennes et al. (2013) discussed this problem, and noted the need for further taxonomic work in this tribe.

Similarity with Sagittaria (line 600): The observations of similar gynoecium and carpel development to Sagittaria (Alismataceae) and the "embedding" of carpels within the receptacle are interesting, because extragynoecial fertilization (via the receptacle) has been reported in both Triuridaceae (Lacandonia) and Sagittaria (for discussion see Rudall et al. 2016).

Other potentially relevant papers on Triuridaceae: It seems surprising that Hemsley (1907) is not cited for his comments on Sciaphila and Seychellaria. Rudall et al. (2016) noted his statement that some specimens of Sciaphila are ``quasi-hermaphrodite'' and often irregular
as to the number and location of the floral parts. It might also be interesting to compare floral orientation and variation in an African triurid, Kupea, which has monosymmetric flowers and considerable variation in organ number (Rudall et al. 2007).
[Hemsley 1907 Annals of Botany 21: 71-77. Rudall et al. 2007, Kew Bulletin 62: 282–292).

·

Basic reporting

The article reports on flower structure and development in poorly studied genus Sciaphila using scanning electron microscopy. Investigation of flower structure and development is sometimes very challenging, especially in plants with very small flowers, as Sciaphila is. Furthermore, Sciaphila plants are achlorophyllous, thus their vegetative parts are reduced making flowers and fruits almost only organs useful for assessment of morphological difference between individual species. As many structures in Sciaphila flowers have been viewed differently by various authors, this study could have significant impact in taxonomy of the genus.
As I am not a native speaker, I do not comment on English. As far as I can judge, the paper is written clearly and professionally. Its structure and content conform to professional standards and the paper offers an insightful introduction to the studied problematics.

Experimental design

The paper presents original primary research. Research questions are clearly stated covering gaps in current knowledge. Methods are described appropriately. Authors used scanning electron microscopy, which was essential to investigate satisfactorily the floral development of Sciaphila flowers. Results are presented as a series of commented SEM photographs, which are of high technical quality. Although all images are relevant to the content of the article, I would welcome descriptions of some SEM photographs to be more detailed. E.g., in Fig 30B no details are given; see also note attached to Fig. 18. While style (its position etc.) is frequently discussed in the text, it is never labelled in presented photographs.

Validity of the findings

The results of the study contribute significantly to the knowledge of floral development and flower structure in the genus Sciaphila. The findings are undoubtedly valid, although stronger results would be obtained if more species (representing a wider range of morphological variation within the genus) were included in the study. However, I agree with all the conclusions stated by the authors.

Additional comments

My only concern is the taxonomical identity of plants used in the study and named as Sciaphila nana. This species was described from Java and the original description did not include male plants, which are usually essential for proper delimitation of Sciaphila species. Therefore, Sciaphila nana is an incompletely known species and naming various populations across Southeast and East Asia by this name is not desirable. This paper is an example of ambiguity connected with this name. Authors used two morphologically clearly different populations named as S. nana and speculated if they even can represent a single species. I would recommend avoiding using this name, or at least use it in the form of Sciaphila cf. nana and explain why the plants are named by this name.

---

## Round 0.2 · accepted · Accept

Thank you for revising your manuscript based on the input from the reviewers.